# Sample Complexity of CVaR Based Risk Sensitive Policy Learning

## Abstract

The conventional offline bandit policy learning literature aims to find a policy that performs well in terms of the average policy effect (APE) on the population, i.e. the *social welfare*. However, in many settings, including healthcare and public policies, the decision-maker also concerns about the *risk* of implementing certain policy. The optimal policy that maximizes social welfare could have a risk of negative effect on some percentage of the worst-affected population, hence not the ideal policy. In this paper, we investigate risk sensitive offline policy learning and its sample complexity, with conditional value at risk (CVaR) of covariate-conditional average policy effect (CAPE) as the risk measure. To this end, we first provide a doubly-robust estimator for the CVaR of CAPE, and show that the this estimator enjoys asymptotic normality even if the nuisance parameters suffer a slower-than-$n^{-\frac{1}{2}}$ estimation rate ($n$ being the sample size). We then propose a risk sensitive learning algorithm that finds the policy maximizing the weighted sum of APE and CVaR of CAPE, within a given policy class $\Pi$. We show that the sample complexity of the proposed algorithm is of the order $O(\kappa(\Pi)n^{-\frac{1}{2}})$, where $\kappa(\Pi)$ is the entropy integral of $\Pi$ under the Hamming distance. The proposed methods are evaluated empirically, demonstrating that by sacrificing not much of the social welfare, our methodology improves the outcome of the worst-affected minority population.

## 1 Introduction

In a variety of fields, more and more decision-makers are learning to target products, services, and information provision based on the user characteristics observed through user-specific historical data (Bertsimas & Kallus, 2020; Bastani & Bayati, 2020; Farias & Li, 2019). For instance, precision medicine learns the optimal personalized treatment from health care records (Kim et al., 2011; Chan et al., 2012; Ozanne et al., 2014); personalized education selects which lessons and learning tools to offer a student on the basis of characteristics and past performance (Tetzlaff et al., 2021); public policies decides personal treatment, e.g. college financial-aid package distribution, re-employment service, etc.(Athey, 2017). These practical needs drive a line of *offline policy learning* literature that is devoted to developing efficient treatment assignment (policy) learning algorithms using historical data (Dudík et al., 2011; Zhang et al., 2012; Swaminathan & Joachims, 2015a;b;c; Kitagawa & Tetenov, 2018; Athey & Wager, 2021; Zhou et al., 2023; Zhan et al., 2023). The optimization objective of most of these works is to maximize the average policy effect (APE) on the population, i.e., the *social welfare*, a key metric in offline policy learning (Rubin, 1974; Zhou et al., 2023).

However, it is widely recognized that policy effects can vary widely between individuals with different characteristics (or covariates in offline policy learning literature), which is a common theme underlying offline policy learning, known as heterogeneity (Crump et al., 2008; Heckman et al., 1997). Therefore, even if the APE on the population is positive, there is a *risk* that many individuals are harmed by the policy employment. Consequently, only considering the population APE does not capture this risk. In many settings discussed previously, besides social welfare, decision-makers concern about the policy effect on the worst-affected population. For example, late stage cancer treatment concentrates on the average treatment effect on the population as well as the worst-possible outcome; education plan considers its impact on the worst-performing students; and government formulating policies would care for negative experience of the worst-affected population. If the risks associated with the policy outweigh the social welfare it generates, deployment of such a policy is not

justifiable to a rational decision-maker who considers equity beyond social welfare, even if the policy is optimal in maximizing social welfare. This calls for a *risk sensitive* policy learning methodology that would improve the outcome of the worst-affected population, and ideally not comprising too much in terms of social welfare.

One appealing resolution is to focus on the distribution of the individual policy effect (IPE), instead of the APE (i.e. the average of IPE over the population) as in the conventional offline policy learning literature. Specifically, the risk sensitive learning object seeks to reduce the policy effect on the worst-affected population, which is the tail of the IPE distribution. A suitable measure for describing this risk is the conditional value at risk (CVaR) of the IPE distribution (Rockafellar et al., 2000), which is the average effects among, say, $\alpha\%$ of the worst-affected population ($\alpha \in [0, 1)$). Hence the risk of the policy performance on the worst-affected $\alpha\%$ of the population can be described by the CVaR of IPE, and risk sensitive policy learning aims to maximize the CVaR of IPE.

One challenge is that the counterfactual IPE of any given policy cannot be directly observed from the observational data. In consequence, it is difficult to learn the distribution of the IPE. However, given rich and continuous covariate spaces, there are well-developed machine learning methods which can be used to estimate covariate-conditional average policy effect (CAPE), which is the expected policy effect conditioned on the individual covariate and would predict IPE well (Künzel et al., 2019; Nie & Wager, 2021; Wager & Athey, 2018). A detailed discussion on CVaR of IPE and CAPE is given in Section 2.1.

Adopting CVaR of CAPE as a policy risk measure, this work aims to fill in the gap between the current offline policy learning literature and the practical needs of risk sensitive policy learning. We present a risk sensitive policy learning algorithm that finds the policy that maximizes the weighted sum of the APE and the CVaR of CAPE, within a given policy class, taking both risk and social welfare into consideration.

### 1.1 OUR CONTRIBUTIONS

**Policy CVaR Inference** Given a policy, we describe the risk of it through CVaR and investigate the relation between the CVaR of IPE and that of CAPE. We provide a doubly robust estimator for CVaR of CAPE, which achieves asymptotic normality even if the nuisance parameters suffer a slower-than-$n^{-\frac{1}{2}}$ estimation rate.

**CVaR based Risk Sensitive Policy Learning** We propose a risk sensitive policy learning scheme that maximizes the weighted sum of APE and CVaR of CAPE over a given policy class $\Pi$. We provide a sample complexity analysis, and show that our algorithm has a suboptimality gap of the order $O(\kappa(\Pi)n^{-\frac{1}{2}})$, where $\kappa(\Pi)$ is a measure quantifying the policy class complexity and $n$ is the number of samples. This result agrees with the sample complexity of other offline policy learning algorithms that maximize social welfare in literature.

**Empirics** We provide efficient implementation of our risk sensitive learning algorithm, and compare its empirical performance with existing benchmark of CAIPWL (Zhou et al., 2023), which aims to maximize the APE. The results present empirical evidence that our risk sensitive policy improves the outcome of the worst-affected population with little compromise in social welfare.

### 1.2 RELATED WORKS

**Risk and CVaR** CVaR is a very popular choice of risk measure, particularly in the finance literature. Various methodologies for the modeling risks through CVaR can be found in Duffie & Pan (1997); Jorion (1996); Pritsker (1997); Morgan (1995); Simons (1996); Beder (1995); Stambaugh (1996); Artzner (1997); Artzner et al. (1999). We refer the readers to Mausser (1998); Embrechts et al. (1999); Pflug (2000) for detailed discussions on CVaR and its properties. Embrechts et al. (1997) provides case studies of CVaR as a risk measure in insurance industry; while Bucay & Rosen (1999); Andersson et al. (2001) used CVaR for credit risk evaluations. Later, Kallus (2023; 2022) used CVaR as a risk measure of treatment effect and discussed inference method of treatment effect CVaR.

**CVaR in Reinforcement Learning** The *reinforcement learning* (RL) literature has pioneered methodologies of risk sensitive learning under a CVaR objective, in the framework of *Markov decision process* (MDP) (Metelli et al., 2021; Sakhi et al., 2024; Behnamnia et al.), where the algorithm learns

while acts (Chow et al., 2015). These works usually assume that propensity score (the probability of choosing an action conditioned on the covariates) is known and Monte Carlo estimation is feasible. In contrast, our setting relies solely on an offline observational data with unknown propensity score, rendering sampling-based methods inapplicable.

More closely related to our work is the literature on risk-sensitive *online and offline policy learning*. Popular multi-armed bandit (MAB) algorithms, such as upper confidence bound and Thompson sampling, have been studied extensively in the context of CVaR based risk sensitive MAB (Galichet, 2015; Galichet et al., 2013; Cassel et al., 2018; Tamkin et al., 2019; Baudry et al., 2021; Tan & Weng, 2023). However, nearly all of these works disregard individual covariates, and thus the resulting algorithms cannot minimize risk at the population level. Qi et al. (2023) studied a similar CVaR based risk minimizing offline policy learning, under the assumption of known behavior policy in the two-action setting, but proved a suboptimal regret bound of $O(n^{-\frac{w}{2w+1}})$ where $w \in (0, 1]$.

**Offline policy learning** There is a long list of works devoted to offline policy learning (Dudík et al., 2011; Zhang et al., 2012; Swaminathan & Joachims, 2015a;b;c; Kitagawa & Tetenov, 2018; Athey & Wager, 2021; Zhou et al., 2023; Zhan et al., 2023; Jin et al., 2021; 2022; Ben-Michael et al., 2024). In particular, Swaminathan & Joachims (2015a) proposed the classical inverse-propensity weight learning (IPWL) that optimizes policy to maximize the APE with known propensity score. Zhou et al. (2023) later introduced the cross-fitted augmented inverse propensity weighted learning (CAIPWL) for learning with unknown propensity score. Policy learning under biased samples and distributional shifts also found to be closely related to CVaR (Sahoo et al., 2022; Lei et al., 2023; Mo et al., 2021).

## 2 PRELIMINARIES

Let $\mathcal{A}$ be the set of $M$ actions $\mathcal{A} := \{1, \cdots, M\}$, and let $\mathcal{X} \subset \mathbb{R}^d$ be a compact set of covariates. Given some action $a \in \mathcal{A}$, the reward distribution $Y(a) \in \mathcal{Y}_a \subset \mathbb{R}$ denotes the potential reward obtained from taking the action $a$. We consider a training dataset $\mathcal{D} = \{(X_i, A_i, Y_i)\}_{i \in [n]}$ consisting of $n$ i.i.d. draws of $(X, A, Y)$ generated as follows.[1] The covariate and potential rewards $(X, Y(1), \cdots, Y(M))$ are drawn from the underlying environment $P$.[2] Some unknown *behavior policy* $\pi_0$ selects an action given the covariate: $A \sim \pi_0(X)$, where the *propensity score* $\pi_0(a \mid X)$ is the probability of $A = a$ given the covariate $X$. In the data set $\mathcal{D}$, only the factual reward corresponding to the chosen action $Y = Y(A)$ is observed. We assume the following for $\pi_0$ and $P$.

**Assumption 2.1** (Regularity)**.** *The behavior policy $\pi_0$ and the environment $P$ satisfy the following: 1. Consistency: $Y = Y(A)$; 2. Unconfoundedness: $(Y(1), \cdots, Y(M)) \perp\!\!\!\perp A | X$; 3. Overlap: for some $\varepsilon > 0$, $\pi_0(a \mid x) \geq \varepsilon$, for all $(a, x) \in \mathcal{A} \times \mathcal{X}$; 4. Bounded Reward: $0 \leq Y(a) \leq \bar{y}$ for $a \in \mathcal{A}$.*

Assumption 2.1 is standard in offline policy learning literature (see e.g., Athey & Wager, 2021; Zhou et al., 2023). The unconfoundedness assumption guarantees identifiability; whiles the overlap assumption ensures sufficient exploration when collecting the data set $\mathcal{D}$ via a positive lower bound on the propensity score. The third assumption of bounded reward support is largely technical to make later analysis tractable. In fact, our methodology can be extended to sub-Gaussian rewards straightforwardly , which we show empirically in Section 5.

Our task is to learn a *risk sensitive policy* $\pi$ in a given policy class $\Pi$ from the training dataset $\mathcal{D}$.

### 2.1 POLICY CONDITIONAL VALUE AT RISK

The policy risk measure of interest is the Conditional Value at Risk (CVaR), which is defined below.

**Definition 2.2** (CVaR)**.** [3] *With respect to a specified probability level $\alpha \in [0, 1]$, the $\alpha$-level Value at Risk (VaR) of a random variable $R \in \mathbb{R}$ is the lowest amount $\beta$ such that, with probability $\alpha$, $R$ will not exceed $\beta$. The $\alpha$-level Conditional Value at Risk (CVaR) is*

$$CVaR_\alpha(R) := \sup_\beta \left( \beta + \frac{1}{\alpha} \mathbb{E}\big[(R - \beta)^-\big] \right). \tag{1}$$

---

[1] We will later use the shorthand $Z := (X, A, Y)$.

[2] Throughout the paper, the expectation $\mathbb{E}$ and probability $\mathbb{P}$ are taken over $P$ unless stated otherwise.

[3] CVaR is sometimes defined for the right tail of $R$, corresponding to $-CVaR(-R)$ in our definition.

**Remark 2.3.** *The sup is attained by $\beta$ being the $\alpha$-quantile: $F_R^{-1}(\alpha) = \inf\{\beta : F_R(\beta) \geq \alpha\}$, where $F_R(r) = \mathbb{P}(R \leq r)$. Here $\beta$ is the $\alpha$-level VaR of R. If R is continuous, then $CVaR_\alpha(R) = \mathbb{E}[R \mid R \leq F_R^{-1}(\alpha)]$; otherwise $CVaR_\alpha(R) \in [\mathbb{E}[R \mid R < F_R^{-1}(\alpha)], \mathbb{E}[R \mid R \leq F_R^{-1}(\alpha)]]$.*

According to Definition 2.2, given a policy $\pi$, the $\alpha$-level CVaR of the IPE $CVaR_\alpha(Y(\pi(X)))$ is the average policy effect among the $(100 \times \alpha)\%$-worst affected population. Let $\mu_\pi(X) := \mathbb{E}[Y(\pi(X)) \mid X]$ denote the CAPE. The next corollary following (Kallus, 2023, Theorem 3.1) gives an upper bound of CVaR of IPE by that of CAPE $CVaR_\alpha(\mu_\pi(X))$.

**Corollary 2.4.** *For any $\alpha \in [0, 1]$ and a policy $\pi$, $CVaR_\alpha(Y(\pi(X))) \leq CVaR_\alpha(\mu_\pi(X))$.*

Since CAPE represents our best guess for IPE, it is reasonable to impute the random and unknown IPE $Y(\pi(X))$ with CAPE $\mu_\pi(X)$. Consequently, $CVaR(\mu_\pi(X))$ can be seen as a substitute for $CVaR(Y(X))$, and a reasonable measure of policy risk.

Formally, our goal is to learn a risk sensitive policy with a high $CVaR_\alpha(\mu_\pi(X))$ from $\mathcal{D}$, with a given target $\alpha$-level. Our challenge is two-fold: (i) inference of $CVaR_\alpha(\mu_\pi(X))$ of a given policy $\pi$ under slow parameter estimation rates of the nuisance parameters; (ii) risk sensitive policy learning whose $\alpha$-level $CVaR_\alpha(\mu_\pi(X))$ is high. Specially, we focus on deriving fast rate policy CVaR estimation and subsequently provide parametric rate sample complexity for policy learning.

## 3 POLICY CVAR INFERENCE

In this section, we concentrate on the first task of policy CVaR inference. We define the policy CVaR

$$\mathcal{V}_\alpha(\pi) := CVaR_\alpha(\mu_\pi(X)) = \sup_\beta \left\{ \beta + \frac{1}{\alpha} \mathbb{E}\big[\big(\mu_\pi(X) - \beta\big)^-\big] \right\}, \tag{2}$$

and denote $\beta_\pi$ as the optimizer $\beta_\pi := \arg\sup_\beta\{\beta + \frac{1}{\alpha}\mathbb{E}[(\mu_\pi(X) - \beta)^-]\}$ in equation 2, which is the $\alpha$-level VaR of $\mu_\pi(X)$.

Since the CAPE $\mu_\pi$ is not directly observed, the first step is fitting it. Let $\hat{\mu}_\pi$ be the estimator of $\mu_\pi$ and let $W_\pi(X_i) := \mathbb{1}\{A_i = \pi(X_i)\}Y_i$. The causal inference literature provides that $\hat{\mu}_\pi$ can be fitted via off-the-shelf estimation algorithms using $\{W_\pi(X_i) : i \in \mathcal{D}\}$ (Hastie et al., 2017; Zhou et al., 2023), e.g., logistic regression, random forests (Ho et al., 1995), kernel regression (Nadaraya, 1964; Watson, 1964), local polynomial regression (Cleveland, 1979; Cleveland & Devlin, 1988).

Given an estimator $\hat{\mu}_\pi$, an naïve policy CVaR estimator is the plug-in estimator

$$\hat{\mathcal{V}}_\alpha^{\text{plug-in}}(\pi) = \sup_\beta (\beta + \frac{1}{n\alpha} \sum_{i \in \mathcal{D}} (\hat{\mu}_\pi(X_i) - \beta)^-).$$

However, the performance of $\hat{\mathcal{V}}_\alpha^{\text{plug-in}}$ depends on the estimation of $\hat{\mu}_\pi$, which is prone to slow convergence rates and potential bias in regression estimation.

We circumvent the issue via a *debiasing* approach (Kallus, 2023) that is insensitive to the estimation of $\mu_\pi$, and thus achieving satisfying policy CVaR estimation rate even in face of the slow convergence rate of $\hat{\mu}_\pi$. Algorithm 1 summaries the inference procedure, which computes the sample average of

$$\phi(\pi, Z; \hat{\pi}_0, \hat{\mu}_\pi, \hat{\beta}_\pi) := \hat{\beta}_\pi + \frac{1}{\alpha} \mathbb{1}\{\hat{\mu}_\pi(X) \leq \hat{\beta}_\pi\} \Big( \hat{\mu}_\pi(X) + \frac{\mathbb{1}\{A = \pi(X)\}}{\hat{\pi}_0(A \mid X)}(Y - \hat{\mu}_\pi(X)) - \hat{\beta}_\pi \Big).$$

Here the propensity estimator $\hat{\pi}_0$ is the estimated propensity score and the estimated policy VaR is

$$\hat{\beta}_\pi = \inf \left\{ \beta : \sum_{i \in \mathcal{D}} (\mathbb{1}\{\hat{\mu}_\pi(X_i) \leq \beta\} - \alpha) \geq 0 \right\}. \tag{3}$$

We also adopt the *cross-fitting* technique (Schick, 1986; Zheng & van der Laan, 2011) over $K$ folds so that the nuisance estimators $(\hat{\mu}_\pi, \hat{\pi}_0, \hat{\beta}_\pi)$ are independent of the data points used for the overall sample average of $\phi$. We split the dataset $\mathcal{D}$ randomly into $K$ fold and denote each fold as $\mathcal{D}^{(k)}$ for $k \in [K]$. At every $k \in [K]$ fold, we use the off fold dataset $\bar{\mathcal{D}}^{(k)} := \{\mathcal{D}^{(i)} : i \not\equiv k \mod K\}$ to estimate the propensity score $\hat{\pi}_0^{(k)}$. Denote $\bar{\mathcal{D}}_\pi^{(k)} := \{(X_i, A_i, Y_i) : i \in \bar{\mathcal{D}}^{(k)}, A_i = \pi(X_i)\}$. We fit

---

**Algorithm 1** Policy CVaR Inference

---

**Input**: Data $\mathcal{D}$, policy $\pi$, CVaR threshold $\alpha$, regression algorithm $\mathcal{R}$ for estimating $\mu_\pi$ and propensity score $\pi_0$.
Randomly split $\mathcal{D}$ into $K$ equally-sized folds;
**for** $k = 1, \cdots, K$ **do**
    Estimate $\hat{\pi}_0^{(k)} \sim \mathcal{R}(\{(X_i, A_i) : i \in \bar{\mathcal{D}}^{(k)}\})$ and $\hat{\mu}_\pi^{(k)} \sim \mathcal{R}(\{(X_i, W_\pi(X_i)) : i \in \bar{\mathcal{D}}_\pi^{(k)}\})$;
    Find $\hat{\beta}_\pi^{(k)}$ with $\hat{\mu}_\pi^{(k)}$ and $\bar{\mathcal{D}}^{(k)}$ as in equation 3;
    Compute the $k$th-fold $\hat{\mathcal{V}}_\alpha^{(k)}(\pi) \leftarrow \frac{1}{|\mathcal{D}^{(k)}|} \sum_{i \in \mathcal{D}^{(k)}} \phi(\pi, Z_i; \hat{\pi}_0^{(k)}, \hat{\mu}_\pi^{(k)}, \hat{\beta}_\pi^{(k)})$;
**end for**
**Output**: $\hat{\mathcal{V}}_\alpha(\pi) = \frac{1}{K} \sum_{k=1}^{K} \hat{\mathcal{V}}_\alpha^{(k)}(\pi)$.

---

$\hat{\mu}_\pi^{(k)}$ by the off fold $\{W_\pi(X_i) : i \in \bar{\mathcal{D}}_\pi^{(k)}\}$. The $k$th fold policy VaR $\hat{\beta}_\pi^{(k)}$ is found via equation 3. Finally, the $k$th fold CVaR estimator is the sample average of $\phi(\pi, Z_i; \hat{\pi}_0^{(k)}, \hat{\mu}_\pi^{(k)}, \hat{\beta}_\pi^{(k)})$ on the $k$th fold $\mathcal{D}^{(k)}$, and the policy CVaR estimator is the sample average of $\{\hat{\mathcal{V}}_\alpha^{(k)}(\pi)\}_{k \in [K]}$.

**Remark 3.1.** *If $\alpha = 1$, then $CVaR_\alpha(\mu_\pi(X)) = \mathbb{E}[\mu_\pi(X)] = \mathbb{E}[Y(\pi(X))]$, and $\hat{\mathcal{V}}_\alpha$ is reduced to the Cross-fitted Augmented Inverse Propensity Weighted (CAIPW) estimator Zhou et al. (2023) for the inference of APE $\mathbb{E}[Y(\pi(X))]$, with unknown propensity scores.*

### 3.1 CONSISTENT POLICY CVAR ESTIMATOR

In this section, we look at the asymptotic behavior of the proposed policy CVaR estimator. We first make some standard assumptions on the estimation rates (Zhou et al., 2023; Kallus, 2023).

**Assumption 3.2** (Asymptotic estimation rate). *Suppose that for each fold $k \in [K]$ and any policy $\pi \in \Pi$, we assume that $\|\hat{\pi}_0^{(k)} - \pi_0\|_{L_2(P)} = o_p(1)$, $\|\hat{\mu}_\pi^{(k)} - \mu_\pi\|_{L_2(P)} = o_p(1)$. Furthermore, we assume that $\|\hat{\pi}_0^{(k)} - \pi_0\|_{L_2(P)} \cdot \|\hat{\mu}_\pi^{(k)} - \mu_\pi\|_{L_2(P)} = o_p(n^{-\frac{1}{2}})$, $\|\hat{\mu}_\pi^{(k)} - \mu_\pi\|_{L_\infty} = o_p(n^{-\frac{1}{4}})$.*

Assumption 3.2 is nonrestrictive, as it suffices to have slow $o_p(n^{-\frac{1}{4}})$-rates on both CAPE and propensity score estimation or no rate on CAPE estimation if the propensity score is known. We impose smoothness of $\mu_\pi$ on the rich covariate space $\mathcal{X}$ to ensure that the CAPE estimator attains the $o_p(n^{-\frac{1}{4}})$ convergence rate in $L_\infty$-norm (Stone, 1982). Recalling the definition of $\mu_\pi$, the smoothness of $\mu_\pi$ is justified as long as the conditional expectation $\mathbb{E}[Y(a) \mid X]$ is well-behaved and smooth in $\mathcal{X}$, which is a common requirement in off-policy learning literature (Zhou et al., 2023). Provided that $\mu_\pi$ is sufficiently smooth, many estimation methods discussed previously in the double-machine-learning estimation literature(Chernozhukov et al., 2018; Farrell, 2015) can easily achieve Assumption 3.2.

We also need another assumption that prohibits degeneracy of the quantile.

**Assumption 3.3** (Regularity of Quantile). *For all $\pi \in \Pi$, we assume that the CDF $F_{\mu_\pi(X)}$ is continuously differentiable at $F_{\mu_\pi(X)}^{-1}(\alpha)$ for the given $\alpha \in [0, 1]$.*

Under well-behaved conditional outcome distributions $\mathbb{P}_{Y(a)|X}, a \in \mathcal{A}$, we may safely assume the above condition holds uniformly for all $\pi \in \Pi$. In particular, we require a locally smooth PDF of $F_{\mu_\pi(X)}'$ around $F_{\mu_\pi}^{-1}(\alpha)$. If $\mu_\pi(X)$ is discrete, Assumption 3.3 can be replaced by $F_{\mu_\pi(X)}^{-1}(\alpha - \epsilon) = F_{\mu_\pi(X)}^{-1}(\alpha + \epsilon)$ for some $\epsilon > 0$ (Kallus, 2023). A sufficient condition is that the CDF $F_{Y(a)|X}$ of the conditional outcome $\mathbb{P}_{Y(a)|X}$ is smooth for $a \in \mathcal{A}$, and, by the bounded-reward assumption, its corresponding PDF is bounded. Under the unconfoundedness assumption in Assumption 2.1, this implies that the induced distribution of $\mu_\pi(X)$ also has a well-behaved PDF. Consequently, we are able to define the uniform bound $\bar{F}_\alpha := \sup_{\pi \in \Pi} F_{\mu_\pi(X)}'(F_{\mu_\pi}^{-1}(\alpha))$ over the policy class $\Pi$.

Since $\hat{\beta}_\pi^{(k)}$ is derived by $\hat{\mu}_\pi^{(k)}$ in equation 3, the following lemma translates the convergence rate of $\hat{\mu}_\pi^{(k)}$ in Assumption 3.2 to that of $\hat{\beta}_\pi^{(k)}$. Its proof is in Appendix E.2.

**Lemma 3.4** (Convergence rate of $\hat{\beta}_\pi$). *Under Assumption 2.1, 3.2 and 3.3, for all $k \in [K]$, the estimation error $|\hat{\beta}_\pi^{(k)} - \beta_\pi| = O_p(n^{-\frac{1}{2}} \vee \|\hat{\mu}_\pi^{(k)} - \mu_\pi\|_{L_r(P)}^{\frac{r}{r+1}}), \forall r \in [1, \infty]$.*

We are now ready to show the the asymptotic normality of the CVaR policy estimator in Algorithm 1, despite of the slow estimation rates in Assumption 3.2. The proof is deferred to Appendix E.3.

**Theorem 3.5** (Asymptotic Normality). *Under Assumption 2.1, 3.2 and 3.3, for any $\pi \in \Pi$, we have* $\sqrt{n}(\hat{\mathcal{V}}_\alpha(\pi) - \mathcal{V}_\alpha(\pi)) \to \mathcal{N}(0, \sigma_\pi^2)$, *where* $\sigma_\pi^2 = Var(\phi(Z; \pi_0, \mu_\pi, \beta_\pi))$.

# 4 CVaR Based Risk Sensitive Policy Learning

We now turn to the second goal and present our CVaR based risk sensitive policy learning ($\lambda$-$\alpha$RSL).

## 4.1 Weighted Policy Value

A straight forward candidate of risk sensitive policy in a policy class $\Pi$ is the one that maximizes the policy CVaR$_\alpha(\mu_\pi(X))$. In many applications, only considering the CVaR objective could be too conservative, as it is also important to monitor the APE. We propose the learning objective that maximizes the policy value $\mathcal{U}_{\lambda,\alpha}(\pi)$, which is the weighted sum of the APE and the policy CVaR with weighting parameter $\lambda \in [0, 1]$:

$$\mathcal{U}_{\lambda,\alpha}(\pi) := \lambda \mathcal{Q}(\pi) + (1 - \lambda)\mathcal{V}_\alpha(\pi), \quad \forall \pi \in \Pi \tag{4}$$

where $\mathcal{Q}(\pi) := \mathbb{E}[Y(\pi(X))] = \mathbb{E}[\mathbb{E}[Y(\pi(X)) \mid X]]$. Detailed discussions of the choice of $\lambda$ empirically and theoretically are given in Section 5 and Appendix B respectively. Zhou et al. (2023) provided the well-known CAIPW Learning (CAIPWL) scheme for policy learning under the APE maximization objective.

We define the optimal policy of a policy class $\Pi$ to be $\pi^* = \max_{\pi \in \Pi} \mathcal{U}_{\lambda,\alpha}(\pi)$. Policy learning task finds a near-optimal robust policy $\pi \in \Pi$ whose policy value is close to the optimal policy. The performance of a learnt policy $\hat{\pi}$ is measured by the sub-optimality gap (regret), defined as

$$R_{\lambda,\alpha}(\hat{\pi}) := \mathcal{U}_{\lambda,\alpha}(\pi^*) - \mathcal{U}_{\lambda,\alpha}(\hat{\pi}). \tag{5}$$

## 4.2 Risk Sensitive Policy Learning

To find the optimal policy $\pi^*$ that maximize the policy value $\mathcal{U}_{\lambda,\alpha}$, the major challenge is the estimations of $\mu_\pi$ and $\beta_\pi$. This is because both $\mu_\pi, \beta_\pi$ are functions of $\pi$, and it is infeasible to estimate for every $\pi$ within a policy class $\Pi$ containing an infinite number of policies.

To tackle the first issue of $\mu_\pi$ estimation, we can express $\mu_\pi(X)$ as a function of the policy action $\pi(X)$: $\mu_\pi(X) = \sum_{a=1}^M \mathbb{1}\{\pi(X) = a\}\mu_a(X)$. To be more precise, we estimate $\mu_a(X)$ by collecting $\{W_a(X_i) = \mathbb{1}\{A_i = a\}Y_i, i \in \mathcal{D}\}_{a \in \mathcal{A}}$. We can construct $\hat{\mu}_\pi(X) = \hat{\mu}_{\pi(X)}(X)$ with $\{\hat{\mu}_a, a \in \mathcal{A}\}$, for any policy $\pi \in \Pi,$. As before, we adopt the cross-fitting technique over $K$ folds to avoid dependence between $\hat{\mu}_\pi$ and the data points used for calculating the sample average.

Deriving the estimator $\hat{\mu}_\pi$ also benefits the learning of the APE $\mathcal{Q}(\pi)$. As discussed before, $\mathcal{Q}(\pi)$ can be learnt via CAIPWL Zhou et al. (2023), which maximizes the CAIPW estimator $\hat{\mathcal{Q}}(\pi)$

$$\psi(\pi, Z; \hat{\pi}_0^{(k)}, \hat{\mu}_\pi^{(k)}) := \frac{\mathbb{1}\{A = \pi(X)\}}{\hat{\pi}_0^{(k)}(\pi(X) \mid X)}(Y - \hat{\mu}_\pi^{(k)}(X)) + \hat{\mu}_\pi^{(k)}(X),$$

$$\hat{\mathcal{Q}}^{(k)}(\pi) = \frac{1}{|\mathcal{D}^{(k)}|} \sum_{i \in \mathcal{D}^{(k)}} \psi(\pi, Z_i; \hat{\pi}_0^{(k)}, \hat{\mu}_\pi^{(k)}), \quad \hat{\mathcal{Q}}(\pi) = \frac{1}{K} \sum_{k=1}^K \hat{\mathcal{Q}}^{(k)}(\pi). \tag{6}$$

Given $\{\hat{\mu}_\pi^{(k)}(X_i)\}_{i \in \bar{\mathcal{D}}^{(k)}}$, equation 3 finds the policy VaR $\hat{\beta}_\pi^{(k)}$ for a specific policy $\pi \in \Pi$. Previously $\mu_\pi(\cdot)$ can be decoupled on the action level, thus transforming the infeasible task of computing a class of infinite nuisance parameters $\{\mu_\pi : \pi \in \Pi\}$ to the feasible task of computing a finite one, however this is not implementable for estimating $\beta_\pi$, which imposes the second challenge. We tackle the issue by jointly optimizing the nuisance parameter $\hat{\beta}_\pi$ and policy $\pi$ (by taking policy gradient updates) in an alternating fashion. In particular, we start by initiating a random policy $\hat{\pi}$ and estimate its $\beta_{\hat{\pi}}$. Then, we take gradient steps to maximize $\hat{\pi} \in \arg\max_{\pi \in \Pi} \hat{\mathcal{U}}_{\lambda,\alpha}(\pi) := \lambda \hat{\mathcal{Q}}(\pi) + (1 - \lambda)\hat{\mathcal{V}}_\alpha(\pi)$, where $\hat{\mathcal{V}}_\alpha(\pi) = \frac{1}{K} \sum_{k=1}^K \sum_{i \in \mathcal{D}^{(k)}} \phi(\pi, Z_i; \hat{\pi}_0^{(k)}, \hat{\mu}_\pi^{(k)}, \hat{\beta}_\pi^{(k)})$, while updating $\beta_{\hat{\pi}}$ along the way. Such process ends when the learnt policy converges. Details of $\lambda$-$\alpha$RSL is in Algorithm 2.

---

**Algorithm 2** $\lambda$-$\alpha$Risk-Sensitive Learning ($\lambda$-$\alpha$RSL)

---

**Input:** Data $\mathcal{D}$, policy class $\Pi$, CVaR threshold $\alpha$, objective weighting parameter $\lambda$, regression algorithm $\mathcal{R}$ for estimating $\mu_a(X)$ and propensity score $\pi_0$.

Randomly split $\mathcal{D}$ into $K$ equally-sized folds;

**for** $k = 1, \cdots, K$ **do**

    Estimate $\hat{\pi}_0^{(k)} \sim \mathcal{R}(\{(X_i, A_i) : i \in \bar{\mathcal{D}}^{(k)}\})$;

    **for** $a \in \mathcal{A}$ **do**

        Estimate $\hat{\mu}_a^{(k)} \sim \mathcal{R}(\{(X_i, W_a(X_i)) : i \in \bar{\mathcal{D}}^{(k)})$;

    **end for**

**end for**

Initiate some $\hat{\pi} \in \Pi$ and estimate $\{\hat{\beta}_{\hat{\pi}}^{(k)}\}_{k \in [K]}$ with $\{\hat{\mu}_a^{(k)}\}_{k \in [K], a \in \mathcal{A}}$;

**while** $\hat{\pi}$ does not converge **do**

    Update $\hat{\pi}$ by some gradient steps to maximize $\hat{\mathcal{U}}_{\lambda,\alpha}(\pi)$;

    Estimate $\{\hat{\beta}_{\hat{\pi}}^{(k)}\}_{k \in [K]}$ with $\{\hat{\mu}_a^{(k)}\}_{k \in [K], a \in \mathcal{A}}$;

**end while**

**Output:** $\hat{\pi}$.

---

**Remark 4.1** (Convergence of $\lambda$-$\alpha$RSL). *We note that the policy learning objective $\hat{\mathcal{U}}_{\lambda,\alpha}(\pi)$ is nonsmooth (due to the indicator functions) with weak concavity structure, which poses particular computation challenges that are common in RL in general (Kaelbling et al., 1996). As the scope of this work does not include developing optimization method for nonsmooth and nonconcave objectives, we defer further discussions on the theoretical convergence of $\lambda$-$\alpha$RSL to Appendix C. The alternating optimization scheme shown in Algorithm 2 is an empirically proven heuristic optimization of $\hat{\mathcal{U}}_{\lambda,\alpha}(\pi)$ that is easy to implement with a variety of optimization methods, including AdaGrad (Duchi et al., 2011) and RMSProp (Hinton et al., 2012; Graves, 2013; Ziyin et al., 2020). We shall see an efficient implementation with a softmax policy class in Section 5.*

### 4.3 Main Regret Analysis

In this section, we present the regret analysis of $\lambda$-$\alpha$RSL. Before we embark on the regret result, we need to introduce the *Hamming entropy integral* $\kappa(\Pi)$, which measures the complexity of $\Pi$.

**Definition 4.2** (Hamming entropy integral). *Given a policy class $\Pi$ and dataset $\{x_1, \ldots, x_n\} \subseteq \mathcal{X}$, (1) the* Hamming distance *between $\pi, \pi' \in \Pi$ as $D_H(\pi, \pi') := \frac{1}{n} \sum_{i=1}^n \mathbb{1}\{\pi(x_i) \neq \pi'(x_i)\}$; (2) the $\epsilon$-covering number of $\{x_1, \ldots, x_n\}$, denoted as $N_H(\epsilon, \Pi; \{x_1, \ldots, x_n\})$, is the smallest number $N$ of policies $\{\pi_1, \ldots, \pi_N\}$ in $\Pi$, such that $\forall \pi \in \Pi$, $\exists \pi'_\ell$ such that $D_H(\pi, \pi_\ell) \leq \epsilon$; (3) the* Hamming entropy integral *of $\Pi$ is defined as $\kappa(\Pi) := \int_0^1 \sqrt{\log N_H(\epsilon^2, \Pi)} \, d\epsilon$, where $N_H(\epsilon, \Pi) := \sup_{n \geq 1} \sup_{x_1, \ldots, x_n} N_H(\epsilon, \Pi; \{x_1, \ldots, x_n\})$.*

We now present the regret guarantee of the policy $\hat{\pi}$ learnt by $\lambda$-$\alpha$RSL. The proof is deferred to Appendix E.4. The main idea is to first decompose the regret

$$R_{\lambda,\alpha}(\hat{\pi}) = \mathcal{U}_{\lambda,\alpha}(\pi^*) - \mathcal{U}_{\lambda,\alpha}(\hat{\pi}) = \lambda(\mathcal{Q}(\pi^*) - \mathcal{Q}(\hat{\pi})) + (1 - \lambda)(\mathcal{V}_\alpha(\pi^*) - \mathcal{V}_\alpha(\hat{\pi})). \quad (7)$$

Note that the first term can be translate to the supremum of the estimation error:

$$\lambda\big(\mathcal{Q}(\pi^*) - \hat{\mathcal{Q}}(\pi^*) + \hat{\mathcal{Q}}(\pi^*) - \hat{\mathcal{Q}}(\hat{\pi}) + \hat{\mathcal{Q}}(\hat{\pi}) - \mathcal{Q}(\hat{\pi})\big) \leq 2\lambda \sup_{\pi \in \Pi} |\mathcal{Q}(\pi) - \hat{\mathcal{Q}}(\pi)|,$$

and bounded by the known results from Zhou et al. (2023). We concentrate on the second term, which can be similarly upper bounded by

$$(1 - \lambda)\big(\mathcal{V}_\alpha(\pi^*) - \mathcal{V}_\alpha(\hat{\pi})\big) \leq 2(1 - \lambda) \sup_{\pi \in \Pi} |\mathcal{V}_\alpha(\pi) - \hat{\mathcal{V}}_\alpha(\pi)|. \quad (8)$$

At a high level, we bound the right hand side of equation 8 by establishing uniform convergence results for the policy CVaR estimators, through a careful chaining argument.

**Theorem 4.3.** *Under Assumption 2.1, 3.2 and 3.3, there exists some $N \in \mathbb{Z}_+$ such that with $n \geq N$ and denoting $q := \sup_{\pi_1, \pi_2 \in \Pi} \mathbb{E}[(\psi(\pi_1, Z; \pi_0, \mu_a) - \psi(\pi_2, Z; \pi_0, \mu_a))^2]$, we have that with*

*probability at least $1 - \Delta$, the regret of $\lambda$-$\alpha$RSL*

$$R_{\lambda,\alpha}(\hat{\pi}) \leq \lambda\sqrt{\frac{q}{n}}\Big(54.4\sqrt{2}\kappa(\Pi) + 435.2 + \sqrt{2\log(1/\Delta)}\Big)$$

$$+ (1-\lambda)\frac{56\bar{y}}{\alpha\varepsilon\sqrt{n}}\Big((8+\alpha\varepsilon)\kappa(\Pi) + \bar{F}_\alpha/3 + (64+5\alpha\varepsilon) + \sqrt{\log(1/\Delta)}\Big).$$

Theorem 4.3 shows that the dependence of $R_{\lambda,\alpha}(\hat{\pi})$ on the sample size $n$ is of order $O(n^{-\frac{1}{2}})$, which agrees with the regret guarantee of CAIPWL Zhou et al. (2023). This implies that the CVaR based risk sensitive policy learning with $\lambda$-$\alpha$RSL attains the same order of sample complexity as other offline policy learning algorithms, especially CAIPWL which maximizes average policy effect, i.e. social welfare, with no consideration of risks.

## 5 EXPERIMENTS

We evaluated the performance of $\lambda$-$\alpha$RSL against the benchmark CAIPWL Zhou et al. (2023).

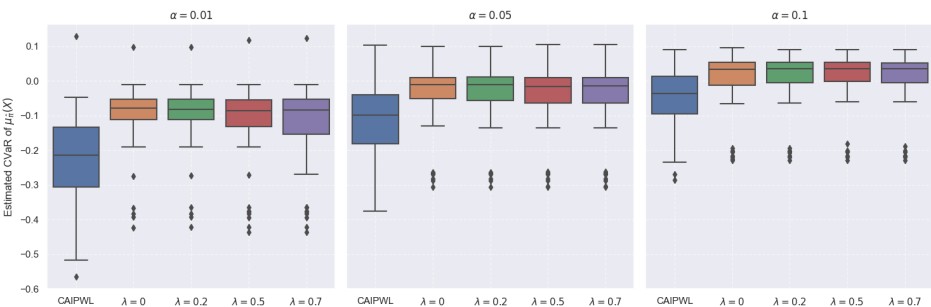

Figure 1: The estimated $\widetilde{\text{CVaR}}_\alpha(\mu_{\hat{\pi}}(X))$ under $\alpha$-level 0.01, 0.05 and 0.1, of learnt policies $\hat{\pi} \in \{\text{CAIPWL}, \lambda\text{-}\alpha\text{RS}, \lambda = 0, 0.2, 0.5, 0.7\}$ on $n = 1000$ training data points, over 50 seeds.

**Data Generating Process** The data generating process follows that of the classical linear boundary example in Si et al. (2023). We generate 50 training datasets of data tuple $(X, A, Y)$, with a behavior policy $\pi_0$; and similarly generate 50 testing datasets, each of size 10,000. The covariate set $\mathcal{X} = \{x \in \mathbb{R}^5 : \|x\|_2 \leq 1\}$ is the closed unit ball of $\mathbb{R}^5$ and the action space is $\mathcal{A} = [3]$. The covariate are sampled independently $X \sim \text{Unif}(\mathcal{X})$; the action $A \sim \pi_0(X)$ and the rewards $Y(a)$'s are mutually independent conditioned on $X$ with $Y(a) \mid X \sim \mathcal{N}(\beta_a^\top X, \sigma_a^2)$, for $\beta_a \in \mathbb{R}^5, \sigma_a \in \mathbb{R}, a \in \mathcal{A}$. Note that the reward distributions here are not of bounded supports.

**Implementation with Softmax Policies** We implement $\lambda$-$\alpha$RSL and the benchmark CAIPWL on a softmax policy class $\Pi$. Given a covariate $x \in \mathcal{X}$, each policy $\pi \in \Pi$ chooses its action $a \in \mathcal{A}$ with probability $\pi(a \mid x) \propto \exp(x^\top \gamma_\pi^a)$ with some policy weights $\{\gamma_\pi^a\}_{a \in \mathcal{A}}$. We consider the neural network softmax policies with a hidden layer of 32 neurons and ReLU activation.

In our implementation, the learning parameters are set to be the same for both $\lambda$-$\alpha$RSL and CAIPWL. The number of data splits is taken to be $K = 2$. We use the Random Forest regressor from the scikit-learn Python library to estimate $\pi_0$ and $\{\mu_a\}_{a \in \mathcal{A}}$. For the policy gradient step, we implement $\lambda$-$\alpha$RSL by maximizing the objective in equation 4 using AdaGrad with a learning rate of 0.01. For CAIPWL, we use RMSProp to maximize its objective equation 6. Since the objective equation 4 and equation 6 are non-convex in the policy weights, following Dudík et al. (2011); Kallus et al. (2022), every policy update is repeated 10 times with perturbed starting weights and the best weights based on the chosen policy learning objective. The policy convergence criteria is whenever the difference between the previous and the updated policy value to be less then 1e-5.

**Performance Metrics** We compare the performance of the learnt policy $\hat{\pi}$ by $\lambda$-$\alpha$RSL and the benchmark CAIPWL with the following two metrics: (i) empirical CVaR of CAPE (empirical policy CVaR); and (ii) empirical APE, on the testing dataset. The two metrics are defined formally as

$$\widetilde{\text{CVaR}}_\alpha(\mu_{\hat{\pi}}(X)) := \hat{\mathbb{E}}_{\mathcal{D}_{\text{test}}}\big[\mu_{\hat{\pi}}(X) \mid \mu_{\hat{\pi}}(X) \leq \hat{F}_{\mu_{\hat{\pi}}(X)}^{-1}(\alpha)\big], \quad \hat{\mathbb{E}}[Y(\hat{\pi}(X))] := \hat{\mathbb{E}}_{\mathcal{D}_{\text{test}}}\big[Y(\hat{\pi}(X))\big].$$

Table 1: The estimated $\tilde{\mathbb{E}}[Y(\hat{\pi}(X))]$ under $\alpha$-level 0.01, 0.05 and 0.1, of learnt policies $\hat{\pi} \in$ {CAIPWL, $\lambda$-$\alpha$RS, $\lambda = 0, 0.2, 0.5, 0.7$} on $n = 1000$ training data points, over 50 seeds.

| $\hat{\pi}$ | $\tilde{\mathbb{E}}[Y(\hat{\pi}(X))]$ | | |
|---|---|---|---|
| $\lambda$-$\alpha$RS | $\alpha = 0.01$ | $\alpha = 0.05$ | $\alpha = 0.1$ |
| $\lambda = 0.0$ | 0.370±1e-2 | 0.365±1e-2 | 0.365±1e-2 |
| $\lambda = 0.2$ | 0.370±1e-2 | 0.365±1e-2 | 0.368±1e-2 |
| $\lambda = 0.5$ | 0.372±1e-2 | 0.368±1e-2 | 0.371±1e-2 |
| $\lambda = 0.7$ | 0.370±1e-2 | 0.368±1e-2 | 0.373±1e-2 |
| CAIPWL | | 0.376±1e-2 | |

Here we use $\hat{F}_Z$ to denote the empirical CDF of a random variable $Z$. For every experiment environment, we test weighting parameters $\lambda \in \{0, 0.2, 0.5, 0.7\}$ and CAIPWL. Note that when $\lambda = 0$, the training objective equation 4 reduces to policy CVaR maximization objective equation 2; when $\lambda = 1$, $\lambda$-$\alpha$RSL reduced to the benchmark CAIPWL.

**Results and Discussion** Figure 1 and Table 1 respectively report the empirical policy CVaR and APE of the learnt policies on training datasets of size $n = 1000$, under $\alpha$-levels 0.01, 0.05 and 0.1. Appendix D provides detailed results of the performances on different samples sizes in Figure 3; and presents the empirical policy CVaRs under large $\alpha$-levels ($\alpha = 0.2, 0.5, 0.9$) in Figure 3.

Our proposed $\lambda$-$\alpha$RSL outperforms the benchmark CAIPWL in terms of empirical policy CVaR, particularly in the small $\alpha$ regime ($\alpha = 0.01, 0.05, 0.1$). On the other hand, CAIPWL attains a higher average empirical APE, whereas $\lambda$-$\alpha$RSL exhibits a modest, yet statistically insignificant, reduction in average empirical APE. For a larger value of $\alpha = 0.2$, although the improvement in policy CVaR provided by $\lambda$-$\alpha$RS diminishes, the quantile of the policy CVaR becomes tighter. This indicates that $\lambda$-$\alpha$RS offers more stable performance with respect to the CVaR criterion. As the value of $\alpha$ increases (particularly when $\alpha \geq 0.5$), the performance of $\lambda$-$\alpha$RS becomes increasingly similar to that of CAIPWL. This occurs because $\lambda$-$\alpha$RS places greater emphasis on the majority of the population as $\alpha$ grows, effectively reducing its objective to that of CAIPWL. Consequently, when $\alpha$ is large, practitioners are primarily optimizing outcomes for the majority group, which is aligned with the goal of CAIPWL, and therefore CAIPWL is recommended in such settings. This also demonstrates that $\lambda$-$\alpha$RSL performs best when the goal is to target and improve the risk, i.e., the worst outcomes experienced by minority groups in the population.

The heatmap of Figure 5 further visualizes this trade-off between risk and social welfare through the weighting parameter $\lambda$ and $\alpha$-level. Large $\lambda$ under small $\alpha$-level results in a much greater improvements in policy CVaR ($\sim$0.12), compared to its loss in APE ($\sim$0.008). This improvement diminishes as $\alpha$ increases. Conversely, a large $\lambda$ helps prevent reductions in social welfare. As $\lambda, \alpha$ both increases, the performance of $\lambda$-$\alpha$RSL is similar to that of CAIPWL.

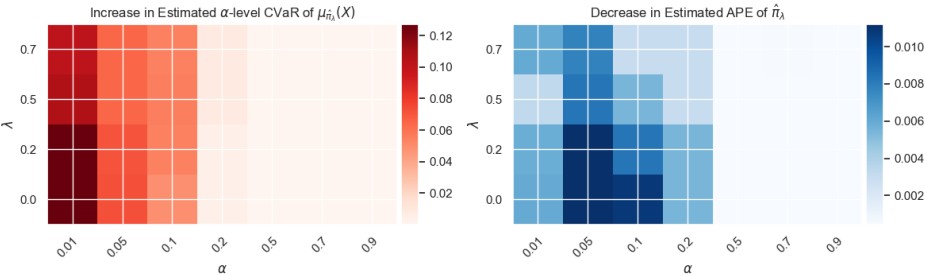

Figure 2: The average $\widetilde{\mathrm{CVaR}}_\alpha(\mu_{\hat{\pi}}(X))$ increase (right) and the average $\tilde{\mathbb{E}}[Y(\hat{\pi}(X))]$ decrease (left) compared to CAIPWL under different $\alpha$-levels and $\lambda$ for $\hat{\pi} = \lambda$-$\alpha$RSL, over 50 seeds.

In conclusion, the empirics show that $\lambda$-$\alpha$RSL improves the outcome of the worst-affected minority population by sacrificing little social welfare.

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

## A  NOTATION

We use $[n]$ to denote the discrete set $\{1, 2, \cdots, n\}$ for any $n \in \mathbb{Z}$. We use $\mathrm{argmin}$ and $\mathrm{argmax}$ to denote the minimizers and maximizers; if the minimzer or the maximizer cannot be attained, we project it back to the feasible set. We denote $u^- := u \wedge 0 = \min\{u, 0\}$ for $u \in \mathbb{R}$. We denote the usual $p$-norm as $\| \cdot \|_p$. For simplicity, we let $\| \cdot \|$ denote the 2-norm $\| \cdot \|_2$. Denote $P$ to be any probability measure defined on the probability space $(\Omega, \sigma(\Omega), P)$. For any function $f$, we denote the $L_r(P)$-norm of $f$ conventionally as $\|f\|_{L_p(P)} = (\int |f(x)|^p \, dP(x))^{1/p}$ and $\|f\|_{L_\infty} = \sup_{x \in \mathcal{X}} |f(x)|$. We also denote $x^- := x \wedge 0 = \min\{x, 0\}$ for $x \in \mathbb{R}$. For any random variables $X, Y$, we use $X \perp\!\!\!\perp Y$ to denote that $X$ is independent of $Y$. For a random variable/vector $X$, we use $\mathbb{E}_X[\cdot]$ to indicate the expectation taken over the distribution of $X$.

## B  WEIGHTING PARAMETER AND CONSTRAINED POLICY LEARNING

As discussed in Section 5, empirically, the weighting parameter $\lambda$ controls how much Algorithm 2 would like to hedge against the policy CVaR. Higher $\lambda$ results in a lower CVaR of CAPE and higher APE.

Theoretically, we can interpret $\lambda$ as an Lagrangian variable of a risk-constrained policy learning problem. The maximization of the policy learning objective in equation 4 is equivalent to

$$\max_{\pi \in \Pi} \mathcal{Q}(\pi) + \frac{1 - \lambda}{\lambda} \mathcal{V}_\alpha(\pi) =: \max_{\pi \in \Pi} \mathcal{Q}(\pi) + \eta \mathcal{V}_\alpha(\pi), \tag{9}$$

where we set $\eta := \frac{1-\lambda}{\lambda}$. The above is equivalent to the Lagrangian form of the CVaR constrained policy learning problem:

$$\max_{\pi \in \Pi} \quad \mathbb{E}[Y(\pi(X))] \tag{10}$$
$$\text{s.t.} \quad \mathrm{CVaR}(\mu_\pi(X)) \geq c,$$

where $c$ is some risk tolerance threshold determined by the decision maker, that satisfies the following assumption.

**Assumption B.1.** *The feasible set $S_c = \{\pi \in \Pi : CVaR_\alpha(\pi(X)) \geq c\}$ is not empty.*

Let $\mu_a(x) = \mathbb{E}[Y(a) \mid X = x]$. Then, by the definition of $\pi$, we can write

$$\mathbb{E}[Y(\pi(X))] = \int_x \sum_{a \in \mathcal{A}} \pi(a \mid x) \mu_a(x) \, d\mathbb{P}_X.$$

Therefore, for any $\pi_1, \pi_2 \in \Pi$ and $t \in (0, 1)$,

$$\mathbb{E}\big[Y\big((t\pi_1 + (1-t)\pi_2)(X)\big)\big]$$
$$= \int_x \sum_{a \in \mathcal{A}} (t\pi_1(a \mid x) + (1-t)\pi_2(a \mid x)) \mu_a(x) \, d\mathbb{P}_X$$
$$= \int_x \sum_{a \in \mathcal{A}} t\pi_1(a \mid x) \mu_a(x) \, d\mathbb{P}_X + \int_x \sum_{a \in \mathcal{A}} (1-t)\pi_2(a \mid x) \mu_a(x) \, d\mathbb{P}_X$$
$$= t\mathbb{E}\big[Y(\pi_1(X))\big] + (1-t)\mathbb{E}\big[Y(\pi_2(X))\big].$$

Combining the above with Assumption B.1 and the concavity of $\mathrm{CVaR}_\alpha(\pi(X))$ shown in Rockafellar et al. (2000), we conclude that the Slater's condition holds and strong duality holds for the below dual of Problem equation 10:

$$\min_{\eta \geq 0} \max_{\pi \in \Pi} \mathbb{E}[Y(\pi(X))] + \eta\big(\mathrm{CVaR}_\alpha(\pi(X)) - c\big).$$

To solve the risk constrained policy learning problem equation 10 using the training dataset $\mathcal{D}$, solve

$$\min_{\eta \geq 0} \max_{\pi \in \Pi} \quad \frac{1}{K} \sum_{k=1}^{K} \hat{\mathcal{Q}}^{(k)}(\pi) + \frac{\eta}{K} \sum_{k=1}^{K} \hat{\mathcal{V}}_\alpha^{(k)}(\pi)$$

$$\text{s.t.} \quad \hat{\beta}_\pi^{(k)} = \inf \left\{ \beta : \sum_{i \in \mathcal{D}^{(k)}} (\mathbb{1}\{\hat{\mu}_\pi^{(k)}(X_i) \leq \beta\} - \alpha) \geq 0 \right\}, \quad \forall k \in [K], \pi \in \Pi,$$

where $\{\hat{\mathcal{Q}}^{(k)}(\pi), \hat{\mathcal{V}}_\alpha^{(k)}(\pi)\}_{k \in [K], \pi \in \Pi}$ are as defined before. Recent literature has provided efficient algorithms to find min-max-min problems as the above. One could apply a first-order method ProM3 in Tu et al. (2024) to solve the risk constrained policy learning problem.

## C  CONVERGENCE OF $\lambda$-$\alpha$RSL

Recall that the policy learning task requires us to maximize the following *deterministic* objective

$$\hat{\mathcal{U}}_{\lambda,\alpha}(\pi) = \frac{1}{K} \sum_{k=1}^{K} \frac{1}{|\mathcal{D}^{(k)}|} \sum_{i \in \mathcal{D}^{(k)}} \left( \lambda \cdot \psi(\pi, Z_i; \hat{\pi}_0^{(k)}, \{\hat{\mu}_a^{(k)}\}_{a \in [M]}) \right.$$

$$\left. + (1 - \lambda) \cdot \phi(\pi, Z_i; \hat{\pi}_0^{(k)}, \{\hat{\mu}_a^{(k)}\}_{a \in [M]}, \{\hat{\beta}_\pi^{(k)}\}_{k \in [K]})) \right),$$

$$=: f(\pi, \{\hat{\beta}_\pi^{(k)}\})$$

with a bilevel structure

$$\max_{\pi \in \Pi} \quad f(\pi, \{\hat{\beta}_\pi^{(k)}\})$$

$$\text{s.t.} \quad \hat{\beta}_\pi^{(k)} = \inf \left\{ \beta : \sum_{i \in \mathcal{D}^{(k)}} (\mathbb{1}\{\hat{\mu}_\pi^{(k)}(X_i) \leq \beta\} - \alpha) \geq 0 \right\}, \quad \forall k \in [K].$$

The inner-level optimization problem has a closed form solution $\{\hat{\beta}_\pi^{(k)}\}$, which is the empirical $\text{VaR}_\alpha(\hat{\mu}_\pi^{(k)}(X_i))$, i.e., the $\alpha|\mathcal{D}^{(k)}|$-th ordered statistics of $\{\mu_\pi^{(k)}(X_i)\}_{i \in \mathcal{D}^{(k)}}$.

On the other hand, the upper-level objective function $f$ is neither smooth nor convex, which poses particular computational challenges. To overcome this issue, consider the smoothed version $\tilde{f}$ for $f$, which adopts the sigmoid approximation for the indicator function in $\phi$:

$$\tilde{f}(\pi, \{\hat{\beta}^{(k)}\}) := \frac{1}{K} \sum_{k=1}^{K} \frac{1}{|\mathcal{D}^{(k)}|} \sum_{i \in \mathcal{D}^{(k)}} \left( \lambda \cdot \psi(\pi, Z_i; \hat{\pi}_0^{(k)}, \{\hat{\mu}_a^{(k)}\}_{a \in [M]}) \right.$$

$$\left. + (1 - \lambda) \cdot \tilde{\phi}(\pi, Z_i; \hat{\pi}_0^{(k)}, \{\hat{\mu}_a^{(k)}\}_{a \in [M]}, \{\hat{\beta}_\pi^{(k)}\}_{k \in [K]})) \right),$$

with $\sigma(x) = \frac{1}{1+e^{-x}}$, $\tau > 0$ a small constant, and

$$\tilde{\phi}(\pi, Z_i; \hat{\pi}_0^{(k)}, \{\hat{\mu}_a^{(k)}\}_{a \in [M]}, \{\hat{\beta}_\pi^{(k)}\}_{k \in [K]})$$

$$= \hat{\beta}_\pi^{(k)} + \frac{1}{\alpha} \cdot \sigma\left( \frac{\hat{\beta}_\pi^{(k)} - \hat{\mu}_\pi^{(k)}(X_i)}{\tau} \right) \cdot \left( \hat{\mu}_\pi^{(k)}(X_i) + \frac{\pi(A_i \mid X_i)}{\hat{\pi}_0^{(k)}(A_i \mid X_i)} \left( Y_i - \hat{\mu}_\pi^{(k)}(X_i) \right) - \hat{\beta}_\pi^{(k)} \right).$$

In this way, we can apply gradient ascent method under the smooth objective $\tilde{f}$. At each time $t$, we take $\pi_{t+1} = \pi_t + \eta \nabla_\pi \tilde{f}(\pi_t, \{\hat{\beta}_t^{(k)}\})$, where $\eta$ is the step size and we denote $\{\hat{\beta}_t^{(k)}\} = \{\hat{\beta}_{\pi_t}^{(k)}\}$. We then update the correct $\{\hat{\beta}_{t+1}^{(k)}\}$.

It can be shown that under the assumptions on the outcome distribution and the propensity scores, the gradient $\nabla_\pi \tilde{f}$ is upper bounded, and thus $\tilde{f}$ is Lipchitz continuous. Subsequently, our policy learning task reduces to a gradient ascent scheme for a Lipchitz continuous but nonconvex objective function. Following the optimization literature (Ghadimi & Lan, 2013), the solution $\pi_T$ converges to a stationary point with a rate of $O(1/\sqrt{T})$, where $T$ is the iteration number.

One limitation is that due to the weak concavity structure of the objective function, we cannot guarantee convergence to the global maximum. We can reformulate the above bilevel optimization problem as a joint optimization problem

$$\max_{\pi \in \Pi, \{\beta_\pi^{(k)}\}} \quad f(\pi, \{\beta_\pi^{(k)}\}),$$

where the optimal $\beta_\pi^{(k)}$ is achieved at

$$\hat{\beta}_\pi^{(k)} = \inf \left\{ \beta : \sum_{i \in \mathcal{D}^{(k)}} (\mathbb{1}\{\hat{\mu}_\pi^{(k)}(X_i) \leq \beta\} - \alpha) \geq 0 \right\}, \quad \forall k \in [K].$$

Under this formulation, the alternating scheme used in $\lambda$-$\alpha$RSL (Algorithm 2) serves as an implementable heuristic for solving the joint problem. Although we have provided empirical evidence of its effectiveness, the convergence properties of this alternating scheme remain an open question and lie beyond the scope of this paper.

# D    EXPERIMENT DETAILS AND MORE RESULTS

**Simulated Dataset Generation Details** We choose the action set $\mathcal{A} = [3]$. Let $\sigma = \{\sigma_a, a \in \mathcal{A}\} = \{0.2, 0.5, 0.8\}$ and let $\{\beta_a, a \in \mathcal{A}\}$ to be

$$\left\{\beta_1 = (1, 0, 0, 0, 0), \beta_2 = (-1/2, \sqrt{3}/2, 0, 0, 0), \beta_3 = (-1/2, -\sqrt{3}/2, 0, 0, 0)\right\}.$$

The underlying policy $\pi_0$ chooses actions with covariate $x$ according to the following rules:

$$(\pi_0(1\,|\,x), \pi_0(2\,|\,x), \pi_0(3\,|\,x)) = \begin{cases} (0.5, 0.25, 0.25), & \text{if } \arg\max_{i=1,2,3}\{\beta_i^\top x\} = 1, \\ (0.25, 0.5, 0.25), & \text{if } \arg\max_{i=1,2,3}\{\beta_i^\top x\} = 2, \\ (0.25, 0.25, 0.5), & \text{if } \arg\max_{i=1,2,3}\{\beta_i^\top x\} = 3. \end{cases}$$

We generate 50 training datasets of

$$\mathcal{D}_{\text{train}} = \left\{(X_i, A_i = \pi_0(X_i), Y_i(\pi_0(X_i)))\right\}_{i=1}^n,$$

where $X_i$'s are sampled i.i.d. uniformly from the closed unit ball of $\mathbb{R}^5$, $A_i \sim \pi_0(X_i)$, and $Y_i(A_i) \sim \mathcal{N}(\beta_{A_i}^\top X_i, \sigma_{A_i}^2)$. Similarly, we sample 50 testing datasets

$$\mathcal{D}_{\text{test}} = \left\{\left(X_i, (Y_i(1), Y_i(2), Y_i(3)), (\mu_1(X_i), \mu_2(X_i), \mu_3(X_i))\right)\right\}_{i=1}^{10,000},$$

where $\mu_a(X_i) = \beta_a^\top X_i$.

**Implementation Details** In our implementation, the learning parameters are set to be the same for both $\lambda$-$\alpha$RSL and CAIPWL. The number of data splits is taken to be $K = 2$. We use the Random Forest regressor from the `scikit-learn` Python library to estimate $\pi_0$ and $\{\mu_a\}_{a\in\mathcal{A}}$. For the policy gradient step, we implement $\lambda$-$\alpha$RSL by maximizing the objective in equation 4 using RMSProp with a learning rate of 0.01. For the benchmark, we similarly use RMSProp to maximize the CAIPWL objective equation 6. Since the objective equation 4 and equation 6 are non-convex in the policy weights, following Dudík et al. (2011); Kallus et al. (2022), every policy update is repeated 10 times with perturbed starting weights and the best weights based on the chosen policy learning objective. The policy convergence criteria is whenever the difference between the previous and the updated policy value to be less then 1e-6.

**Computation Details** The experiments were run on the following cloud servers: (i) an Intel Xeon Platinum 8160 @ 2.1 GHz with 766GB RAM and 96 CPU x 2.1 GHz; (ii) an Intel Xeon Platinum 8160 @ 2.1 GHz with 1.5TB RAM and 96 CPU x 2.1 GHz; (iii) an Intel Xeon Gold 6132 @ 2.59 GHz with 768GB RAM and 56 CPU x 2.59 GHz and (iv) an Intel Xeon GPU E5-2697A v4 @ 2.59 GHz with 384GB RAM and 64 CPU x 2.59 GHz.

**More Results** We now provide detailed results of the policies' performances on different samples sizes in Figure 3, and the empirical policy CVaRs under large $\alpha$-levels ($\alpha = 0.2, 0.5, 0.9$) in Figure 3. For a value of $\alpha = 0.2$, although the improvement in policy CVaR provided by $\lambda$-$\alpha$RS diminishes, the quantile of the policy CVaR becomes tighter. When $\alpha \geq 0.5$, there is no significant statistical evidence that the performance of $\lambda$-$\alpha$RSL is different from that of CAIPWL.

We also test the policy CVaR inference task. We implement Algorithm 1 on the training dataset and estimate the policy CVaR of a fixed policy $\pi$, which is different from the behavior policy $\pi_0$. The performance of Algorithm 1 is evaluated by the mean squared error (MSE) of the estimated policy CVaR. Figure 5 shows the MSE of the estimated policy CVaR by Algorithm 1, with $\alpha$-level $\{0.1, 0.05, 0.01\}$. A variant of Algorithm 1 with known propensity score is also tested. As the sample size increases, the estimation becomes more accurate and stable. With large sample size, the estimation with unknown propensity score is comparable to the one with known propensity score, which highlights the double-robustness of our estimator. We also observe that with larger $\alpha$, Algorithm 1 needs more samples to achieve small MSE.

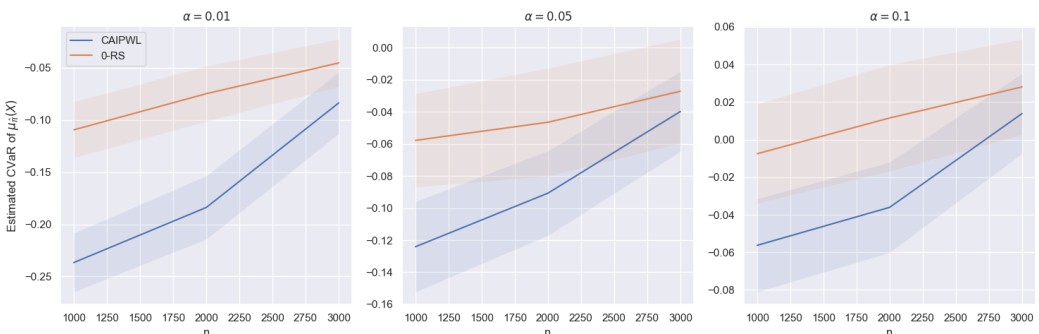

Figure 3: The estimated $\widetilde{\mathrm{CVaR}}_\alpha(\mu_{\hat\pi}(X))$ under $\alpha$-level 0.01, 0.05 and 0.1, over 50 seeds, of learnt policies $\hat\pi \in \{\text{CAIPWL}, \lambda\text{-}\alpha\text{RS}, \lambda = 0, 0.2, 0.5, 0.7\}$ on $n = 1000, 3000, 5000$ training data points.

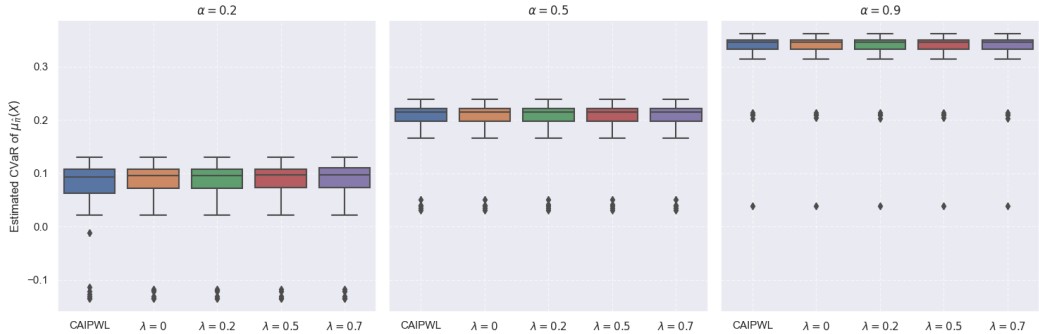

Figure 4: The estimated $\widetilde{\mathrm{CVaR}}_\alpha(\mu_{\hat\pi}(X))$ under $\alpha$-level 0.2, 0.5 and 0.9, over 50 seeds, of learnt policies $\hat\pi \in \{\text{CAIPWL}, \lambda\text{-}\alpha\text{RS}, \lambda = 0, 0.2, 0.5, 0.7\}$ on $n = 1000$ training data points.

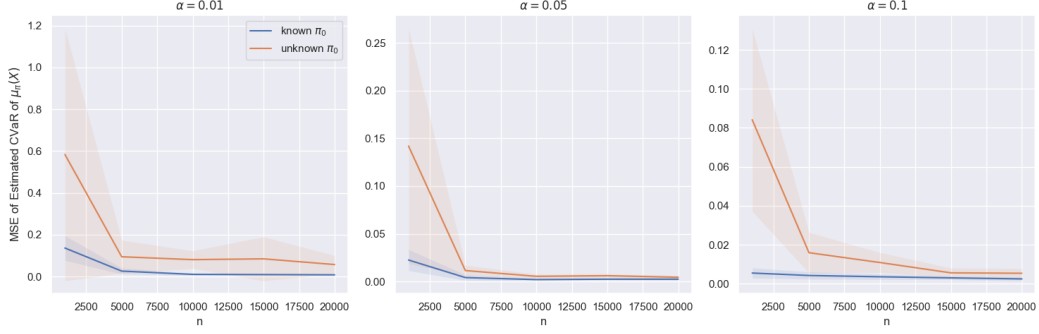

Figure 5: Average MSE of estimated policy CVaR by Algorithm 1 with unknown and known propensity score, over 25 seeds. $\alpha$-level is chosen to be 0.01, 0.05 and 0.1.

# E  DEFERRED PROOFS OF THE MAIN RESULTS

## E.1  PROOF OF COROLLARY 2.4

*Proof.* We follow Theorem 3.1 in Kallus (2023). By Jensen' inequality,

$$\text{CVaR}_\alpha(Y(\pi(X))) = \sup_\beta \left( \beta + \frac{1}{\alpha} \mathbb{E}\left[ \mathbb{E}[(Y(\pi(X)) - \beta)^- \mid X] \right] \right)$$

$$\leq \sup_\beta \left( \beta + \frac{1}{\alpha} \mathbb{E}(\mu_\pi(X) - \beta)^- \right) = \text{CVaR}_\alpha(\mu_\pi(X)).$$

$\square$

## E.2  PROOF OF LEMMA 3.4

*Proof.* Denote the quantile $Q_\alpha(f)$ of any function $f(x)$ as $Q_\alpha(f) = \inf\{\beta : \mathbb{E}[\mathbb{1}\{f(X) \leq \beta\} - \alpha] \geq 0\}$. We also denote the empirical quantile using the $k$th off fold data as

$$\hat{Q}_\alpha^{(k)}(f) = \inf \left\{ \beta : \sum_{i \in \bar{\mathcal{D}}^{(k)}} (\mathbb{1}\{f(X_i) \leq \beta\} - \alpha) \geq 0 \right\}.$$

As in Algorithm 1, we have $\hat{\beta}_\pi^{(k)} = \hat{Q}_\alpha^{(k)}(\hat{\mu}_\pi^{(k)})$, and the true $\beta_\pi = F_{\mu_\pi(X)}^{-1}(\alpha) = Q_\alpha(\mu_\pi)$.

We will show the equality by proving that the RHS is the upper bound and the lower bound of the LHS. We first prove the upper bound of case where $r = \infty$. By definition of $\hat{Q}_\alpha^{(k)}$, we have that

$$|\hat{Q}_\alpha^{(k)}(\hat{\mu}_\pi^{(k)}) - Q_\alpha(\mu_\pi)| \leq \sup_{i \in \bar{\mathcal{D}}^{(k)}} |\hat{\mu}_\pi^{(k)}(X_i) - \mu_\pi(X_i)| = O_p(\|\hat{\mu}_\pi^{(k)} - \mu_\pi\|_{L_\infty}).$$

Now we consider the case where $r < \infty$. Let $\delta = \|\mu_\pi - \hat{\mu}_\pi^{(k)}\|_{L_r(P)}^{\frac{r}{r+1}}$. By a union bound with respect to the empirical distribution,

$$\hat{Q}_\alpha^{(k)}(\hat{\mu}^{(k)}) \leq \hat{Q}_{\alpha+\delta}^{(k)}(\mu_\pi) + \hat{Q}_{1-\delta}^{(k)}(\hat{\mu}_\pi^{(k)} - \mu_\pi).$$

By continuous differentiability in Assumption 3.3, the first term on the RHS can be bounded by

$$\hat{Q}_{\alpha+\delta}^{(k)}(\mu_\pi) = \hat{Q}_{\alpha+\delta}^{(k)}(\mu_\pi) - Q_{\alpha+\delta}(\mu_\pi) + Q_{\alpha+\delta}(\mu_\pi)$$

$$\leq \hat{Q}_{\alpha+\delta}^{(k)}(\mu_\pi) - Q_{\alpha+\delta}(\mu_\pi) + Q_\alpha(\mu_\pi) + O_p(\delta).$$

Furthermore, using the delta method, we have that $\hat{Q}_{\alpha+\delta}^{(k)}(\mu_\pi) - Q_{\alpha+\delta}(\mu_\pi) = O_P(n^{-\frac{1}{2}})$ and,

$$\hat{Q}_{\alpha+\delta}^{(k)}(\mu_\pi) \leq O_p(n^{-\frac{1}{2}}) + Q_\alpha(\mu_\pi) + O_p(\delta).$$

To upper bound the second term, we apply Markov's inequality with respect to the empirical distribution:

$$\hat{Q}_{1-\delta}^{(k)}(\hat{\mu}_\pi^{(k)} - \mu_\pi)$$

$$= \inf \left\{ \beta : \sum_{i \in \bar{\mathcal{D}}^{(k)}} (\mathbb{1}\{\hat{\mu}_\pi^{(k)}(X_i) - \mu_\pi(X_i) \leq \beta\} - (1 - \|\mu_\pi - \hat{\mu}_\pi^{(k)}\|^{\frac{r}{r+1}})) \geq 0 \right\}$$

$$\leq \frac{(\frac{1}{|\bar{\mathcal{D}}^{(k)}|} \sum_{i \in \bar{\mathcal{D}}^{(k)}} |\hat{\mu}_\pi^{(k)}(X_i) - \mu_\pi(X_i)|^r)^{\frac{1}{r}}}{\delta^{-\frac{1}{r}}} = O_p(\delta^{\frac{1}{r}} \|\hat{\mu}_\pi^{(k)} - \mu_\pi\|_{L_r(P)}) = O_p(\delta).$$

Combining the two results, we have

$$\hat{Q}_\alpha^{(k)}(\hat{\mu}^{(k)}) \leq O_p(n^{-\frac{1}{2}}) + Q_\alpha(\mu_\pi) + O_p(\delta)$$

$$\Rightarrow \hat{Q}_\alpha^{(k)}(\hat{\mu}^{(k)}) - Q_\alpha(\mu_\pi) \leq O_p(n^{-\frac{1}{2}}) + O_p(\|\mu_\pi - \hat{\mu}_\pi^{(k)}\|_{L_r(P)}^{\frac{r}{r+1}}).$$

To derive a lower bound, we can make a symmetric argument by similarly writing:

$$\hat{Q}_\alpha^{(k)}(\hat{\mu}^{(k)}) \geq \hat{Q}_{\alpha-\delta}^{(k)}(\mu_\pi) + \hat{Q}_{1+\delta}^{(k)}(\hat{\mu}_\pi^{(k)} - \mu_\pi).$$

The upper bound and lower bound gives the desired result when $r < \infty$.  $\square$

### E.3 PROOF OF THEOREM 3.5

*Proof.* We first state the following helper lemma, the proof of which can be found in Appendix F.1.

**Lemma E.1.** *Suppose that Assumption 2.1, 3.2, and 3.3 hold. Then there exists some constant $c_1 > 0$ such that $\|\mu_\pi - \hat{\mu}\|_{L_\infty} \le c_1$ and $|\beta_\pi - \hat{\beta}| \le c_1$, and for any $\alpha \in (0, 1]$, we have*

$$|\mathbb{E}[\phi(\pi, Z; \hat{\pi}_0, \hat{\mu}, \hat{\beta})] - \mathbb{E}[\phi(\pi, Z; \pi_0, \mu_\pi, \beta_\pi)]|$$

$$\le \frac{2\bar{y}}{\alpha\varepsilon}\|\hat{\pi}_0 - \pi_0\|_{L_2(P)}\|\hat{\mu} - \mu_\pi\|_{L_2(P)}$$

$$+ \frac{1}{\alpha}(F'_{\mu_\pi(X)}(F^{-1}_{\mu_\pi(X)}(\alpha)) + 1)(\|\hat{\mu}_\pi - \mu_\pi\|_{L_\infty} + |\beta_\pi - \hat{\beta}|)^2$$

$$+ \frac{1}{2\alpha}(F'_{\mu_\pi(X)}(F^{-1}_{\mu_\pi(X)}(\alpha)) + 1)|\hat{\beta} - \beta_\pi|^2$$

$$\|\phi(Z; \hat{\pi}_0, \hat{\mu}, \hat{\beta}) - \phi(Z; \pi_0, \mu_\pi, \beta_\pi)\|_{L_2(P)}$$

$$\le \frac{2\bar{y}}{\alpha\varepsilon^{3/2}}\|\hat{\pi}_0 - \pi_0\|_{L_2(P)} + \frac{2}{\alpha\varepsilon}\|\hat{\mu}(X) - \mu_\pi(X)\|_{L_2(P)}$$

$$+ \frac{16\bar{y}}{\alpha\varepsilon}(F'_{\mu_\pi(X)}(F^{-1}_{\mu_\pi(X)}(\alpha)) + 1)(|\hat{\beta} - \beta_\pi| + \|\hat{\mu}(X) - \mu_\pi(X)\|_{L_\infty}).$$

*Fixing a sample $i \in \mathcal{D}^{(k)}$, we also have*

$$|\phi(Z_i; \hat{\pi}_0, \hat{\mu}, \hat{\beta}) - \phi(Z_i; \pi_0, \mu_\pi, \beta_\pi)| \le \frac{2\bar{y}}{\alpha\varepsilon^{3/2}}|\hat{\pi}_0(\pi(X_i) \mid X_i) - \pi_0(\pi(X_i) \mid X_i)| + \frac{1}{\alpha}|\hat{\beta} - \beta_\pi|$$

$$+ \frac{1}{\alpha\varepsilon}|\hat{\mu}(X_i) - \mu_\pi(X_i)| + \frac{7\bar{y}}{\alpha\varepsilon}.$$

In the following sequel, we shall show that $\hat{\mathcal{V}}_\alpha^{(k)} = \mathcal{V}_\alpha(\pi) + o_p(n^{-\frac{1}{2}})$, for all data fold $k \in [K]$.

We can decompose

$$\hat{\mathcal{V}}_\alpha^{(k)} - \mathcal{V}_\alpha(\pi)$$

$$= \frac{1}{|\mathcal{D}^{(k)}|} \sum_{i \in \mathcal{D}^{(k)}} \phi(\pi, Z_i; \hat{\pi}_0^{(k)}, \hat{\mu}_\pi^{(k)}, \hat{\beta}_\pi^{(k)}) - \phi(\pi, Z_i; \pi_0, \mu_\pi, \beta_\pi)$$

$$= \frac{1}{|\mathcal{D}^{(k)}|} \sum_{i \in \mathcal{D}^{(k)}} \phi(\pi, Z_i; \hat{\pi}_0^{(k)}, \hat{\mu}_\pi^{(k)}, \hat{\beta}_\pi^{(k)}) - \mathbb{E}[\phi(\pi, Z; \hat{\pi}_0^{(k)}, \hat{\mu}_\pi^{(k)}, \hat{\beta}_\pi^{(k)}) \mid \bar{\mathcal{D}}^{(k)}] - \phi(\pi, Z_i; \pi_0, \mu_\pi, \beta_\pi)$$

$$+ \mathbb{E}[\phi(\pi, Z; \pi_0, \mu_\pi, \beta_\pi) \mid \bar{\mathcal{D}}^{(k)}] + \mathbb{E}[\phi(\pi, Z; \hat{\pi}_0^{(k)}, \hat{\mu}_\pi^{(k)}, \hat{\beta}_\pi^{(k)}) \mid \bar{\mathcal{D}}^{(k)}] - \mathbb{E}[\phi(\pi, Z; \pi_0, \mu_\pi, \beta_\pi) \mid \bar{\mathcal{D}}^{(k)}]$$

$$= \mathbb{E}[\phi(\pi, Z; \hat{\pi}_0^{(k)}, \hat{\mu}_\pi^{(k)}, \hat{\beta}_\pi^{(k)}) \mid \bar{\mathcal{D}}^{(k)}] - \mathbb{E}[\phi(\pi, Z; \pi_0, \mu_\pi, \beta_\pi) \mid \bar{\mathcal{D}}^{(k)}]$$

$$+ \frac{1}{|\mathcal{D}^{(k)}|} \sum_{i \in \mathcal{D}^{(k)}} \left( \phi(\pi, Z_i; \hat{\pi}_0^{(k)}, \hat{\mu}_\pi^{(k)}, \hat{\beta}_\pi^{(k)}) - \phi(\pi, Z_i; \pi_0, \mu_\pi, \beta_\pi) \right.$$

$$\left. - \left( \mathbb{E}[\phi(\pi, Z; \hat{\pi}_0^{(k)}, \hat{\mu}_\pi^{(k)}, \hat{\beta}_\pi^{(k)}) \mid \bar{\mathcal{D}}^{(k)}] - \mathbb{E}[\phi(\pi, Z; \pi_0, \mu_\pi, \beta_\pi) \mid \bar{\mathcal{D}}^{(k)}] \right) \right) =: (I) + (II).$$

We will show that Term $(I), (II)$ are both $o_p(n^{-\frac{1}{2}})$.

By Lemma E.1 and Lemma 3.4, Term $(I)$ is

$$(I) \le \frac{2\bar{y}}{\alpha\varepsilon}\|\hat{\pi}_0 - \pi_0\|_{L_2(P)}\|\hat{\mu} - \mu_\pi\|_{L_2(P)} + \frac{1}{\alpha}(F'_{\mu_\pi(X)}(F^{-1}_{\mu_\pi(X)}(\alpha)) + 1)(\|\hat{\mu}_\pi - \mu_\pi\|_{L_\infty} + |\beta_\pi - \hat{\beta}|)^2$$

$$+ \frac{1}{2\alpha}(F'_{\mu_\pi(X)}(F^{-1}_{\mu_\pi(X)}(\alpha)) + 1)|\hat{\beta} - \beta_\pi|^2$$

$$= O_p(\|\hat{\pi}_0^{(k)} - \pi_0\|_{L_2(P)}\|\hat{\mu}_\pi^{(k)} - \mu_\pi\|_{L_2(P)} + \|\hat{\mu}_\pi - \mu_\pi\|_{L_\infty}^2 + \|\hat{\mu}_\pi - \mu_\pi\|_{L_\infty}|\hat{\beta}_\pi^{(k)} - \beta_\pi| + |\hat{\beta}_\pi^{(k)} - \beta_\pi|^2)$$

$$= O_p(\|\hat{\pi}_0^{(k)} - \pi_0\|_{L_2(P)}\|\hat{\mu}_\pi^{(k)} - \mu_\pi\|_{L_2(P)} + \|\hat{\mu}_\pi^{(k)} - \mu_\pi\|_{L_\infty}^2).$$

By Assumption 3.2, we have that Term $(I) = o_p(n^{-\frac{1}{2}})$.

Conditioned on the off-fold data $\bar{\mathcal{D}}^{(k)}$, we apply Chebyshev's inequality to Term $(II)$. For any $t > 0$, we have that

$$\mathbb{P}\big(|II| \geq t \mid \bar{\mathcal{D}}^{(k)}\big) \leq \frac{\text{Var}\big(\|\phi(Z; \hat{\pi}_0^{(k)}, \hat{\mu}_\pi^{(k)}, \hat{\beta}_\pi^{(k)}) - \phi(Z; \pi_0, \mu_\pi, \beta_\pi)\|\big)}{|\mathcal{D}^{(k)}|t^2}$$

$$\leq \frac{1}{|\mathcal{D}^{(k)}|t^2}\bigg(\frac{2\bar{y}}{\alpha\varepsilon^{3/2}}\|\hat{\pi}_0^{(k)} - \pi_0\|_{L_2(P)} + |\hat{\beta}_\pi^{(k)} - \beta_\pi|$$

$$+ \frac{16\bar{y}}{\alpha\varepsilon}(F'_{\mu_\pi(X)}(F^{-1}_{\mu_\pi(X)}(\alpha)) + 1)\big(|\hat{\beta}_\pi^{(k)} - \beta_\pi| + \|\hat{\mu}_\pi^{(k)}(X) - \mu_\pi(X)\|_{L_\infty}\big)\bigg),$$

where the last step is due to Lemma E.1. Consequently,

$$(II) = O_p\bigg(\frac{\|\phi(Z; \hat{\pi}_0^{(k)}, \hat{\mu}_\pi^{(k)}, \hat{\beta}_\pi^{(k)}) - \phi(Z; \pi_0, \mu_\pi, \beta_\pi)\|_{L_2(P)}}{|\mathcal{D}^{(k)}|^{\frac{1}{2}}}\bigg).$$

By Lemma 3.4, and Assumption 3.2, we further have

$$(II) = O_p(|\mathcal{D}^{(k)}|^{-\frac{1}{2}}(\|\hat{\pi}_0 - \pi_0\|_{L_2(P)} + \|\hat{\mu} - \mu_\pi\|_{L_2(P)} + |\beta_\pi - \hat{\beta}|))$$

$$= O_p(|\mathcal{D}^{(k)}|^{-\frac{1}{2}}(\|\hat{\pi}_0 - \pi_0\|_{L_2(P)} + \|\hat{\mu} - \mu_\pi\|_{L_2(P)} + n^{-\frac{1}{2}} \vee \|\hat{\mu}_\pi^{(k)} - \mu_\pi\|_{L_2(P)}^{\frac{2}{3}}))$$

$$= O_p(|\mathcal{D}^{(k)}|^{-\frac{1}{2}}o_p(1)) = o_p(n^{-\frac{1}{2}}).$$

We conclude that $\hat{\mathcal{V}}_\alpha^{(k)} = \mathcal{V}_\alpha(\pi) + o_p(n^{-\frac{1}{2}})$, for all data fold $k \in [K]$. Thus

$$\sqrt{n}(\hat{\mathcal{V}}_\alpha - \mathcal{V}_\alpha(\pi)) = \frac{1}{\sqrt{n}}\sum_{i \in \mathcal{D}}\big(\phi(Z_i; \pi_0, \mu_\pi, \beta_\pi) - \mathcal{V}_\alpha(\pi)\big) + o_p(1),$$

and it converges in distribution $\mathcal{N}(0, \sigma_\pi^2)$ by the central limit theorem and Slutsky's theorem. The asymptotic variance is

$$\sigma_\pi^2 = \text{Var}(\phi(Z; \pi_0, \mu_\pi, \beta_\pi)).$$

$\square$

## E.4 PROOF OF THEOREM 4.3

We first write the second term of equation 7, without the $(1 - \lambda)$ scale, as

$$\mathcal{V}_\alpha(\pi^*) - \mathcal{V}_\alpha(\hat{\pi}) = \mathcal{V}_\alpha(\pi^*) - \hat{\mathcal{V}}_\alpha(\pi^*) + \hat{\mathcal{V}}_\alpha(\pi^*) - \hat{\mathcal{V}}_\alpha(\hat{\pi}) + \hat{\mathcal{V}}_\alpha(\hat{\pi}) - \mathcal{V}_\alpha(\hat{\pi})$$

$$\leq 2\sup_{\pi \in \Pi}|\mathcal{V}_\alpha(\pi) - \hat{\mathcal{V}}_\alpha(\pi)| = 2\sup_{\pi \in \Pi}|\mathcal{V}_\alpha(\pi) - \tilde{\mathcal{V}}_\alpha(\pi) + \tilde{\mathcal{V}}_\alpha(\pi) - \hat{\mathcal{V}}_\alpha(\pi)|$$

$$\leq \underbrace{\sup_{\pi \in \Pi}2|\mathcal{V}_\alpha(\pi) - \tilde{\mathcal{V}}_\alpha(\pi)|}_{(1)} + \underbrace{\sup_{\pi \in \Pi}2|\tilde{\mathcal{V}}_\alpha(\pi) - \hat{\mathcal{V}}_\alpha(\pi)|}_{(2)}. \tag{11}$$

We will show the upper bound of both terms separately.

As an important intermediate step, we first establish a regret bound a regret bound for the policy when the algorithm has access to the quantities $\pi_0(x), \mu_a(x)$. Note that when the true $\pi_0, \{\mu_a\}_{a \in \{0,1\}}$ are known, the oracle policy learning CVaR estimator does not rely on cross-fold fitting as it is designed for deriving independent $\hat{\pi}_0, \{\hat{\mu}_a\}_{a \in \{0,1\}}$ estimators. Also note that if we are given $\{\mu_a\}_{a \in \{0,1\}}$, then for every $\pi \in \Pi$, we can find the oracle policy VaR

$$\beta_\pi = \arg\sup_\beta\bigg\{\beta + \frac{1}{\alpha}\mathbb{E}\big[\big(\mu_\pi(X) - \beta\big)^-\big]\bigg\}. \tag{12}$$

We also denote the oracle $\alpha$-level policy CVaR as

$$\tilde{\mathcal{V}}_\alpha = \frac{1}{|\mathcal{D}|}\sum_{i \in \mathcal{D}}\phi(\pi, Z_i; \pi_0, \mu_\pi, \beta_\pi).$$

The following lemma provides the oracle regret of Term (1) in equation 11, and the proof of which can be found in Appendix F.2.

**Lemma E.2.** *Under Assumption 2.1, 3.2 and 3.3, with probability at least $1 - \Delta$,*

$$\sup_{\pi \in \Pi} |\mathcal{V}_\alpha(\pi) - \tilde{\mathcal{V}}_\alpha(\pi)| \leq \frac{16\bar{y}}{\sqrt{n}}(\kappa(\Pi) + 7) + \frac{(12 + \sqrt{2})\bar{y}}{\sqrt{n}} + o\left(\frac{1}{\sqrt{n}}\right).$$

The proof of Theorem 4.3 also utilized the following result, which upper bounds Term (2) in equation 11, and the proof is deferred to Appendix F.3.

**Corollary E.3.** *Under Assumption 2.1, 3.2 and 3.3, there exists some $N \in \mathbb{Z}_+$ such that with $n \geq N$, we have that with probability at least $1 - \Delta$,*

$$\sup_{\pi \in \Pi} |\hat{\mathcal{V}}_\alpha(\pi) - \tilde{\mathcal{V}}_\alpha(\pi)| \leq \frac{28\bar{y}}{\alpha\varepsilon\sqrt{n}}\big(8\kappa(\Pi) + 62 + \sqrt{\log(1/\Delta)}\big) + \frac{2\bar{y} + 9\bar{F}_\alpha}{\alpha\varepsilon\sqrt{n}} + o\left(\frac{1}{\sqrt{n}}\right).$$

*Proof of Theorem 4.3.* By the regret decomposition as in equation 11 and the results from Lemma E.2 and Corollary E.3, there exists some $N \in \mathbb{Z}_+$ such that with $n \geq N$, we have that with probability at least $1 - \Delta$,

$$\mathcal{V}_\alpha(\pi^*) - \mathcal{V}_\alpha(\hat{\pi}) \leq \sup_{\pi \in \Pi} 2|\mathcal{V}_\alpha(\pi) - \tilde{\mathcal{V}}_\alpha(\pi)| + \sup_{\pi \in \Pi} 2|\tilde{\mathcal{V}}_\alpha(\pi) - \hat{\mathcal{V}}_\alpha(\pi)|$$

$$\leq \frac{56\bar{y}}{\alpha\varepsilon\sqrt{n}}\big(8\kappa(\Pi) + \bar{F}_\alpha/3 + 64 + \sqrt{\log(1/\Delta)}\big) + \frac{56\bar{y}}{\sqrt{n}}\big(\kappa(\Pi) + 5\big).$$

The proof concludes by (Zhou et al., 2023, Theorem 3), the above result and the regret decomposition equation 7. □

# F PROOF OF TECHNICAL LEMMAS

## F.1 PROOF OF LEMMA E.1

*Proof.* First, we can compute the expectation

$$\mathbb{E}[\phi(\pi, Z_i; \hat{\pi}_0, \hat{\mu}, \hat{\beta})] = \mathbb{E}\left[\mathbb{E}\left[\hat{\beta} + \frac{1}{\alpha}\mathbb{1}\{\hat{\mu}(X_i) \leq \hat{\beta}\}\left(\hat{\mu}(X_i) + \frac{\mathbb{1}\{A_i = \pi(X_i)\}}{\hat{\pi}_0(\pi(X_i) \mid X_i)}(Y_i - \hat{\mu}(X_i)) - \hat{\beta}\right) \mid X_i\right]\right]$$

$$= \mathbb{E}\left[\hat{\beta} + \frac{1}{\alpha}\mathbb{1}\{\hat{\mu}(X_i) \leq \hat{\beta}\}\left(\hat{\mu}(X_i) + \frac{\pi_0(\pi(X_i) \mid X_i)}{\hat{\pi}_0(\pi(X_i) \mid X_i)}(\mu_\pi(X_i) - \hat{\mu}(X_i)) - \hat{\beta}\right)\right]$$

$$= \hat{\beta} + \frac{1}{\alpha}\mathbb{E}\left[\mathbb{1}\{\hat{\mu}(X) \leq \hat{\beta}\}\left(\hat{\mu}(X) + \frac{\pi_0(\pi(X) \mid X)}{\hat{\pi}_0(\pi(X) \mid X)}(\mu_\pi(X) - \hat{\mu}(X)) - \hat{\beta}\right)\right].$$

$$\tag{13}$$

The first inequality in the statement can be decomposed into the following:

$$|\mathbb{E}[\phi(Z; \hat{\pi}_0, \hat{\mu}, \hat{\beta})] - \mathbb{E}[\phi(Z; \pi_0, \mu_\pi, \beta_\pi)]|$$

$$= |\mathbb{E}[\phi(Z; \hat{\pi}_0, \hat{\mu}, \hat{\beta})] - \mathbb{E}[\phi(Z; \pi_0, \hat{\mu}, \hat{\beta})] + \mathbb{E}[\phi(Z; \pi_0, \hat{\mu}, \hat{\beta})] - \mathbb{E}[\phi(Z; \pi_0, \mu_\pi, \hat{\beta})] + \mathbb{E}[\phi(Z; \pi_0, \mu_\pi, \hat{\beta})]$$

$$- \mathbb{E}[\phi(Z; \pi_0, \mu_\pi, \beta_\pi)]|$$

$$\leq \underbrace{|\mathbb{E}[\phi(Z; \hat{\pi}_0, \hat{\mu}, \hat{\beta})] - \mathbb{E}[\phi(Z; \pi_0, \hat{\mu}, \hat{\beta})]|}_{(I)} + \underbrace{|\mathbb{E}[\phi(Z; \pi_0, \hat{\mu}, \hat{\beta})] - \mathbb{E}[\phi(Z; \pi_0, \mu_\pi, \hat{\beta})]|}_{(II)}$$

$$+ \underbrace{|\mathbb{E}[\phi(Z; \pi_0, \mu_\pi, \hat{\beta})] - \mathbb{E}[\phi(Z; \pi_0, \mu_\pi, \beta_\pi)]|}_{(III)}.$$

We will bound the three terms $(I), (II), (III), (IV)$ individually. We first look at Term $(I)$:

$$(I) = \left| \hat{\beta} + \frac{1}{\alpha}\mathbb{E}\left[ \mathbb{1}\{\hat{\mu}(X) \le \hat{\beta}\} \left( \hat{\mu}(X) + \frac{\pi_0(\pi(X) \mid X)}{\hat{\pi}_0(\pi(X) \mid X)}(\mu_\pi(X) - \hat{\mu}(X)) - \hat{\beta} \right) \right] \right.$$

$$\left. - \hat{\beta} - \frac{1}{\alpha}\mathbb{E}\left[ \mathbb{1}\{\hat{\mu}(X) \le \hat{\beta}\} \left( \hat{\mu}(X) + \frac{\pi_0(\pi(X) \mid X)}{\pi_0(\pi(X) \mid X)}(\mu_\pi(X) - \hat{\mu}(X)) - \hat{\beta} \right) \right] \right|$$

$$= \left| \frac{1}{\alpha}\mathbb{E}\left[ \mathbb{1}\{\hat{\mu}(X) \le \hat{\beta}\} \left( \hat{\mu}(X) + \frac{\pi_0(\pi(X) \mid X)}{\hat{\pi}_0(\pi(X) \mid X)}(\mu_\pi(X) - \hat{\mu}(X)) - \hat{\beta} \right) \right] \right.$$

$$\left. - \frac{1}{\alpha}\mathbb{E}\left[ \mathbb{1}\{\hat{\mu}(X) \le \hat{\beta}\} \left( \hat{\mu}(X) + \frac{\hat{\pi}_0(\pi(X) \mid X)}{\hat{\pi}_0(\pi(X) \mid X)}(\mu_\pi(X) - \hat{\mu}(X)) - \hat{\beta} \right) \right] \right|$$

$$\le \left| \frac{1}{\alpha}\mathbb{E}\left[ \mathbb{1}\{\hat{\mu}(X) \le \hat{\beta}\} \frac{|\pi_0(\pi(X) \mid X) - \hat{\pi}_0(\pi(X) \mid X)|}{\hat{\pi}_0(\pi(X) \mid X)}|\mu_\pi(X) - \hat{\mu}(X)| \right] \right|$$

$$\le \frac{2\bar{y}}{\alpha\varepsilon}\|\hat{\pi}_0 - \pi_0\|_{L_2(P)}\|\hat{\mu} - \mu\|_{L_2(P)}.$$

By continuous density Assumption 3.3, there exists some $c_1 > 0$ such that $\mu_\pi(X) - \beta_\pi$ has a density on $(-3c_1, 3c_1)$ bounded by $F'_{\mu_\pi(X)}(F^{-1}_{\mu_\pi(X)}(\alpha)) + 1$. Therefore, provided that $|\hat{\beta} - \beta_\pi| \le c_1$ and $\|\hat{\mu}(X) - \mu_\pi(X)\|_{L_\infty} \le c_1$,

$$(II) = \left| \hat{\beta} + \frac{1}{\alpha}\mathbb{E}\left[ \mathbb{1}\{\hat{\mu}(X) \le \hat{\beta}\} \left( \hat{\mu}(X) + \frac{\pi_0(\pi(X) \mid X)}{\pi_0(\pi(X) \mid X)}(\mu_\pi(X) - \hat{\mu}(X)) - \hat{\beta} \right) \right] \right.$$

$$\left. - \hat{\beta} - \frac{1}{\alpha}\mathbb{E}\left[ \mathbb{1}\{\mu_\pi(X) \le \hat{\beta}\} \left( \mu_\pi(X) + \frac{\pi_0(\pi(X) \mid X)}{\pi_0(\pi(X) \mid X)}(\mu_\pi(X) - \mu_\pi(X)) - \hat{\beta} \right) \right] \right|$$

$$= \left| \frac{1}{\alpha}\mathbb{E}[(\mu_\pi(X) - \hat{\beta})(\mathbb{1}\{\hat{\mu}(X) \le \hat{\beta}\} - \mathbb{1}\{\mu_\pi(X) \le \hat{\beta}\})] \right|$$

$$= \left| \frac{1}{\alpha}\mathbb{E}\left[ (\mu_\pi(X) - \hat{\beta})\left( \mathbb{1}\{\mu_\pi(X) - \beta_\pi \le \hat{\beta} - \beta_\pi + \mu_\pi(X) - \hat{\mu}(X)\} - \mathbb{1}\{\mu_\pi(X) - \beta_\pi \le \hat{\beta} - \beta_\pi\} \right) \right] \right|$$

$$\le \frac{1}{\alpha}\mathbb{E}\left[ |\mu_\pi(X) - \hat{\beta}|\mathbb{1}\{|\mu_\pi(X) - \beta_\pi| \le |\hat{\beta} - \beta_\pi| + |\hat{\mu}(X) - \mu_\pi(X)|\} \right]$$

$$\le \frac{1}{\alpha}\mathbb{E}\left[ |\mu_\pi(X) - \hat{\beta}|\mathbb{1}\{|\mu_\pi(X) - \beta_\pi| \le |\hat{\beta} - \beta_\pi| + \|\hat{\mu}(X) - \mu_\pi(X)\|_{L_\infty}\} \right]$$

$$\le \frac{1}{\alpha}(F'_{\mu_\pi(X)}(F^{-1}_{\mu_\pi(X)}(\alpha)) + 1)\left( |\hat{\beta} - \beta_\pi| + \|\hat{\mu}(X) - \mu_\pi(X)\|_{L_\infty} \right)^2.$$

Finally we analyze Term $(III)$. Define

$$f(\beta) = \mathbb{E}[\phi(Z; \pi_0, \mu_\pi, \beta)] = \beta + \frac{1}{\alpha}\mathbb{E}[(\mu_\pi(X) - \beta)^-]$$

By definition $f'(\beta_\pi) = 0$ and $|f''(\beta)| \le \frac{1}{\alpha}(F'_{\mu_\pi(X)}(F^{-1}_{\mu_\pi(X)}(\alpha)) + 1)$ for $\beta \in (\beta_\pi - c_1, \beta_\pi + c_1)$.

Therefore, provided with the assumption that $|\hat{\beta} - \beta_\pi| \le c_1/3$, by Taylor's theorem, we can upper bound Term $(III)$ by:

$$(III) \le \frac{1}{2\alpha}(F'_{\mu_\pi(X)}(F^{-1}_{\mu_\pi(X)}(\alpha)) + 1)|\hat{\beta} - \beta_\pi|^2.$$

Now we turn to the second inequality. The difference in interest can be written as

$$\|\phi(Z; \hat{\pi}_0, \hat{\mu}, \hat{\beta}) - \phi(Z; \pi_0, \mu_\pi, \beta_\pi)\|_{L_2(P)}$$

$$= \|\phi(Z; \hat{\pi}_0, \hat{\mu}, \hat{\beta}) - \phi(Z; \pi_0, \hat{\mu}, \hat{\beta}) + \phi(Z; \pi_0, \hat{\mu}, \hat{\beta}) - \phi(Z; \pi_0, \hat{\mu}, \beta_\pi) + \phi(Z; \pi_0, \hat{\mu}, \beta_\pi)$$

$$- \phi(Z; \pi_0, \mu_\pi, \beta_\pi)\|_{L_2(P)}$$

$$\le \underbrace{\|\phi(Z; \hat{\pi}_0, \hat{\mu}, \hat{\beta}) - \phi(Z; \pi_0, \hat{\mu}, \hat{\beta})\|_{L_2(P)}}_{(1)} + \underbrace{\|\phi(Z; \pi_0, \hat{\mu}, \hat{\beta}) - \phi(Z; \pi_0, \hat{\mu}, \beta_\pi)\|_{L_2(P)}}_{(2)}$$

$$+ \underbrace{\|\phi(Z; \pi_0, \hat{\mu}, \beta_\pi) - \phi(Z; \pi_0, \mu_\pi, \beta_\pi)\|_{L_2(P)}}_{(3)}.$$

We will upper bound the three terms above individually. For Term $(1)$, we compute that

$$
\begin{aligned}
(1) =& \left\| \frac{1}{\alpha} \mathbb{1}\{\hat{\mu}(X) \le \hat{\beta}\} \left( \frac{\mathbb{1}\{A = \pi(X)\}}{\hat{\pi}_0(\pi(X) \mid X)} - \frac{\mathbb{1}\{A = \pi(X)\}}{\pi_0(\pi(X) \mid X)} \right) (Y - \hat{\mu}(X)) \right\|_{L_2(P)} \\
=& \left\| \frac{1}{\alpha} \mathbb{1}\{\hat{\mu}(X) \le \hat{\beta}\} \mathbb{1}\{A = \pi(X)\} \left( \frac{1}{\hat{\pi}_0(\pi(X) \mid X)} - \frac{1}{\pi_0(\pi(X) \mid X)} \right) (Y - \hat{\mu}(X)) \right\|_{L_2(P)},
\end{aligned}
$$

where the last equality uses the fact that $\hat{\pi}_0(0 \mid X) + \hat{\pi}_0(1 \mid X) = 1$ and that $\pi_0(0 \mid X) + \pi_0(1 \mid X) = 1$. By Assumption 2.1, we have that

$$
\left\| \frac{1}{\hat{\pi}_0(\pi(X) \mid X)} - \frac{1}{\pi_0(\pi(X) \mid X)} \right\|_{L_2(P)} \le \varepsilon^{-3/2} \|\hat{\pi}_0 - \pi_0\|_{L_2(P)}, \quad \|Y - \hat{\mu}(X)\|_{L_2(P)} \le 2\bar{y},
$$

and thus $(1) \le \frac{2\bar{y}}{\alpha \varepsilon^{3/2}} \|\hat{\pi}_0 - \pi_0\|_{L_2(P)}$.

We also compute Term $(2)$:

$$
\begin{aligned}
(2) =& \left\| (\hat{\beta} - \beta_\pi) + \frac{1}{\alpha} \mathbb{1}\{\hat{\mu}(X) \le \hat{\beta}\} \left( \hat{\mu}(X) + \frac{\mathbb{1}\{A = \pi(X)\}}{\pi_0(\pi(X) \mid X)} (Y - \hat{\mu}(X)) - \hat{\beta} \right) \right. \\
& \left. - \frac{1}{\alpha} \mathbb{1}\{\hat{\mu}(X) \le \beta_\pi\} \left( \hat{\mu}(X) + \frac{\mathbb{1}\{A = \pi(X)\}}{\pi_0(\pi(X) \mid X)} (Y - \hat{\mu}(X)) - \beta_\pi \right) \right\|_{L_2(P)} \\
=& \left\| (\hat{\beta} - \beta_\pi) - \frac{1}{\alpha} \mathbb{1}\{\hat{\mu}(X) \le \hat{\beta}\} \hat{\beta} + \frac{1}{\alpha} \mathbb{1}\{\hat{\mu}(X) \le \beta_\pi\} \beta_\pi + \frac{1}{\alpha} \mathbb{1}\{\hat{\mu}(X) \le \hat{\beta}\} \beta_\pi - \frac{1}{\alpha} \mathbb{1}\{\hat{\mu}(X) \le \hat{\beta}\} \beta_\pi \right. \\
& \left. + \frac{1}{\alpha} (\mathbb{1}\{\hat{\mu}(X) \le \hat{\beta}\} - \mathbb{1}\{\hat{\mu}(X) \le \beta_\pi\}) \left( \hat{\mu}(X) + \frac{\mathbb{1}\{A = \pi(X)\}}{\pi_0(\pi(X) \mid X)} (Y - \hat{\mu}(X)) - \beta_\pi \right) \right\|_{L_2(P)} \\
=& \left\| (\hat{\beta} - \beta_\pi) \left( 1 - \frac{\mathbb{1}\{\hat{\mu}(X) \le \hat{\beta}\}}{\alpha} \right) \right. \\
& \left. + \frac{1}{\alpha} (\mathbb{1}\{\hat{\mu}(X) \le \hat{\beta}\} - \mathbb{1}\{\hat{\mu}(X) \le \beta_\pi\}) \left( \hat{\mu}(X) + \frac{\mathbb{1}\{A = \pi(X)\}}{\pi_0(\pi(X) \mid X)} (Y - \hat{\mu}(X)) - \beta_\pi \right) \right\|_{L_2(P)}.
\end{aligned}
$$

By Assumption 2.1, we have that $|\beta_\pi| \le \bar{y}$. Therefore, Term $(2)$ is bounded by

$$
(2) \le \frac{1}{\alpha} |\hat{\beta} - \beta_\pi| + \frac{4\bar{y}}{\alpha \varepsilon} \| \mathbb{1}\{\hat{\mu}(X) \le \hat{\beta}\} - \mathbb{1}\{\hat{\mu}(X) \le \beta_\pi\} \|_{L_2(P)}.
$$

Now, applying a similar trick as in the analysis of Term $(II)$, we have that

$$
\begin{aligned}
\| \mathbb{1}\{\hat{\mu}(X) \le \hat{\beta}\} - \mathbb{1}\{\hat{\mu}(X) \le \beta_\pi\} \|_{L_2(P)} \le& \mathbb{P}(|\mu(X) - \beta_\pi| \le |\hat{\beta} - \beta_\pi| + \|\hat{\mu}(X) - \mu_\pi(X)\|_{L_\infty}) \\
\le& 2(F'_{\mu_\pi(X)}(F^{-1}_{\mu_\pi(X)}(\alpha)) + 1)(|\hat{\beta} - \beta_\pi| + \|\hat{\mu}(X) - \mu_\pi(X)\|_{L_\infty}),
\end{aligned}
$$

where the last inequality is due to the fact that $|\hat{\beta} - \beta_\pi| \le c/3$, and $\|\hat{\mu} - \mu_\pi\|_{L_\infty} \le c/3$. Finally, Term $(2)$ is upper bounded by

$$
(2) \le \frac{1}{\alpha} |\hat{\beta} - \beta_\pi| + \frac{8\bar{y}}{\alpha \varepsilon} (F'_{\mu_\pi(X)}(F^{-1}_{\mu_\pi(X)}(\alpha)) + 1)(|\hat{\beta} - \beta_\pi| + \|\hat{\mu}(X) - \mu_\pi(X)\|_{L_\infty}).
$$

Similarly, Term (3) can be written as

$$
(3) = \left\| \frac{1}{\alpha} \mathbb{1}\{\hat{\mu}(X) \le \beta_\pi\} \left( \hat{\mu}(X) + \frac{\mathbb{1}\{A = \pi(X)\}}{\pi_0(\pi(X) \mid X)}(Y - \hat{\mu}(X)) - \beta_\pi \right) \right.
$$

$$
\left. - \frac{1}{\alpha} \mathbb{1}\{\mu_\pi(X) \le \beta_\pi\} \left( \mu_\pi(X) + \frac{\mathbb{1}\{A = \pi(X)\}}{\pi_0(\pi(X) \mid X)}(Y - \mu_\pi(X)) - \beta_\pi \right) \right\|_{L_2(P)}
$$

$$
= \left\| \frac{1}{\alpha}(\mathbb{1}\{\hat{\mu}(X) \le \beta_\pi\} - \mathbb{1}\{\mu_\pi(X) \le \beta_\pi\})\left( \frac{\mathbb{1}\{A = \pi(X)\}}{\pi_0(\pi(X) \mid X)}Y - \beta_\pi \right) \right.
$$

$$
+ \frac{1}{\alpha} \mathbb{1}\{\hat{\mu}(X) \le \beta_\pi\}\left( \left(1 + \frac{\mathbb{1}\{A = \pi(X)\}}{\pi_0(\pi(X) \mid X)}\right)\hat{\mu}(X) \right)
$$

$$
\left. - \frac{1}{\alpha} \mathbb{1}\{\mu_\pi(X) \le \beta_\pi\}\left( \left(1 + \frac{\mathbb{1}\{A = \pi(X)\}}{\pi_0(\pi(X) \mid X)}\right)\mu_\pi(X) \right) \right\|_{L_2(P)}
$$

$$
= \left\| \frac{1}{\alpha}(\mathbb{1}\{\hat{\mu}(X) \le \beta_\pi\} - \mathbb{1}\{\mu_\pi(X) \le \beta_\pi\})\left( \frac{\mathbb{1}\{A = \pi(X)\}}{\pi_0(\pi(X) \mid X)}Y - \beta_\pi \right) \right.
$$

$$
+ \frac{1}{\alpha} \mathbb{1}\{\hat{\mu}(X) \le \beta_\pi\}\left( \left(1 + \frac{\mathbb{1}\{A = \pi(X)\}}{\pi_0(\pi(X) \mid X)}\right)\hat{\mu}(X) \right)
$$

$$
- \frac{1}{\alpha} \mathbb{1}\{\mu_\pi(X) \le \beta_\pi\}\left( \left(1 + \frac{\mathbb{1}\{A = \pi(X)\}}{\pi_0(\pi(X) \mid X)}\right)\mu_\pi(X) \right)
$$

$$
+ \frac{1}{\alpha} \mathbb{1}\{\hat{\mu}(X) \le \beta_\pi\}\left( \left(1 + \frac{\mathbb{1}\{A = \pi(X)\}}{\pi_0(\pi(X) \mid X)}\right)\mu_\pi(X) \right)
$$

$$
\left. - \frac{1}{\alpha} \mathbb{1}\{\hat{\mu}(X) \le \beta_\pi\}\left( \left(1 + \frac{\mathbb{1}\{A = \pi(X)\}}{\pi_0(\pi(X) \mid X)}\right)\mu_\pi(X) \right) \right\|_{L_2(P)}.
$$

Rearrange, we have that

$$
(3) \le \underbrace{\left\| \frac{1}{\alpha}(\mathbb{1}\{\hat{\mu}(X) \le \beta_\pi\} - \mathbb{1}\{\mu_\pi(X) \le \beta_\pi\})\left( \frac{\mathbb{1}\{A = \pi(X)\}}{\pi_0(\pi(X) \mid X)}Y - \beta_\pi \right) \right\|_{L_2(P)}}_{(3)_I}
$$

$$
+ \underbrace{\left\| \frac{1}{\alpha} \mathbb{1}\{\hat{\mu}(X) \le \beta_\pi\}\left( 1 - \frac{\mathbb{1}\{A = \pi(X)\}}{\pi_0(\pi(X) \mid X)}\right)(\hat{\mu}(X) - \mu_\pi(X)) \right\|_{L_2(P)}}_{(3)_{II}}
$$

$$
+ \underbrace{\left\| \frac{1}{\alpha}(\mathbb{1}\{\hat{\mu}(X) \le \beta_\pi\} - \mathbb{1}\{\mu_\pi(X) \le \beta_\pi\})\left( 1 - \frac{\mathbb{1}\{A = \pi(X)\}}{\pi_0(\pi(X) \mid X)}\right)\mu_\pi(X) \right\|_{L_2(P)}}_{(3)_{III}}
$$

By the result of Term (2), we have that

$$
\|\mathbb{1}\{\hat{\mu}(X) \le \beta_\pi\} - \mathbb{1}\{\mu_\pi(X) \le \beta_\pi\}\|_{L_2(P)} \le 2(F'_{\mu_\pi(X)}(F^{-1}_{\mu_\pi(X)}(\alpha)) + 1)\|\hat{\mu}(X) - \mu_\pi(X)\|_{L_\infty}.
$$

Therefore, we can bound

$$
(3)_I \le \frac{4\bar{y}}{\alpha\varepsilon}(F'_{\mu_\pi(X)}(F^{-1}_{\mu_\pi(X)}(\alpha)) + 1)\|\hat{\mu}(X) - \mu_\pi(X)\|_{L_\infty}
$$

$$
(3)_{II} \le \frac{2}{\alpha\varepsilon}\|\hat{\mu}(X) - \mu_\pi(X)\|_{L_2(P)}
$$

$$
(3)_{III} \le \frac{4\bar{y}}{\alpha\varepsilon}(F'_{\mu_\pi(X)}(F^{-1}_{\mu_\pi(X)}(\alpha)) + 1)\|\hat{\mu}(X) - \mu_\pi(X)\|_{L_\infty}.
$$

Putting everything together, we have that

$$
\|\phi(Z; \hat{\pi}_0, \hat{\mu}, \hat{\beta}) - \phi(Z; \pi_0, \mu_\pi, \beta_\pi)\|_{L_2(P)}
$$

$$
\le \frac{2\bar{y}}{\alpha\varepsilon^{3/2}}\|\hat{\pi}_0 - \pi_0\|_{L_2(P)} + \frac{2}{\alpha\varepsilon}\|\hat{\mu}(X) - \mu_\pi(X)\|_{L_2(P)}
$$

$$
+ \frac{16\bar{y}}{\alpha\varepsilon}(F'_{\mu_\pi(X)}(F^{-1}_{\mu_\pi(X)}(\alpha)) + 1)\left(|\hat{\beta} - \beta_\pi| + \|\hat{\mu}(X) - \mu_\pi(X)\|_{L_\infty}\right).
$$

For the last inequality, we note that we can similarly decompose

$$|\phi(Z_i; \hat{\pi}_0, \hat{\mu}, \hat{\beta}) - \phi(Z_i; \pi_0, \mu_\pi, \beta_\pi)|$$
$$= |\phi(Z_i; \hat{\pi}_0, \hat{\mu}, \hat{\beta}) - \phi(Z_i; \pi_0, \hat{\mu}, \hat{\beta}) + \phi(Z_i; \pi_0, \hat{\mu}, \hat{\beta}) - \phi(Z_i; \pi_0, \hat{\mu}, \beta_\pi)$$
$$+ \phi(Z_i; \pi_0, \hat{\mu}, \beta_\pi) - \phi(Z_i; \pi_0, \mu_\pi, \beta_\pi)|$$
$$\leq \underbrace{|\phi(Z_i; \hat{\pi}_0, \hat{\mu}, \hat{\beta}) - \phi(Z_i; \pi_0, \hat{\mu}, \hat{\beta})|}_{(1)} + \underbrace{|\phi(Z_i; \pi_0, \hat{\mu}, \hat{\beta}) - \phi(Z_i; \pi_0, \hat{\mu}, \beta_\pi)|}_{(2)}$$
$$+ \underbrace{|\phi(Z_i; \pi_0, \hat{\mu}, \beta_\pi) - \phi(Z_i; \pi_0, \mu_\pi, \beta_\pi)|}_{(3)}.$$

We will upper bound the three terms above individually. For Term (1), we compute that

$$(1) = \left| \frac{1}{\alpha} \mathbb{1}\{\hat{\mu}(X_i) \leq \hat{\beta}\} \left( \frac{\mathbb{1}\{A_i = \pi(X_i)\}}{\hat{\pi}_0(\pi(X_i) \mid X_i)} - \frac{\mathbb{1}\{A_i = \pi(X_i)\}}{\pi_0(\pi(X_i) \mid X_i)} \right) (Y_i - \hat{\mu}(X_i)) \right|$$
$$= \left| \frac{1}{\alpha} \mathbb{1}\{\hat{\mu}(X_i) \leq \hat{\beta}\} \mathbb{1}\{A_i = \pi(X_i)\} \left( \frac{1}{\hat{\pi}_0(\pi(X_i) \mid X_i)} - \frac{1}{\pi_0(\pi(X_i) \mid X_i)} \right) (Y_i - \hat{\mu}(X_i)) \right|,$$

where the last equality uses the fact that $\hat{\pi}_0(0 \mid X_i) + \hat{\pi}_0(1 \mid X_i) = 1$ and that $\pi_0(0 \mid X_i) + \pi_0(1 \mid X_i) = 1$. Since

$$\left| \frac{1}{\hat{\pi}_0(\pi(X_i) \mid X_i)} - \frac{1}{\pi_0(\pi(X_i) \mid X_i)} \right| \leq \varepsilon^{-3/2} |\hat{\pi}_0(\pi(X_i) \mid X_i) - \pi_0(\pi(X_i) \mid X_i)|, \quad |Y_i - \hat{\mu}(X_i)| \leq 2\bar{y},$$

we have that $(1) \leq \frac{2\bar{y}}{\alpha \varepsilon^{3/2}} |\hat{\pi}_0(\pi(X_i) \mid X_i) - \pi_0(\pi(X_i) \mid X_i)|$.

We also compute Term (2):

$$(2) = \left| (\hat{\beta} - \beta_\pi) + \frac{1}{\alpha} \mathbb{1}\{\hat{\mu}(X_i) \leq \hat{\beta}\} \left( \hat{\mu}(X_i) + \frac{\mathbb{1}\{A_i = \pi(X_i)\}}{\pi_0(\pi(X_i) \mid X_i)} (Y_i - \hat{\mu}(X_i)) - \hat{\beta} \right) \right.$$
$$\left. - \frac{1}{\alpha} \mathbb{1}\{\hat{\mu}(X_i) \leq \beta_\pi\} \left( \hat{\mu}(X_i) + \frac{\mathbb{1}\{A_i = \pi(X_i)\}}{\pi_0(\pi(X_i) \mid X_i)} (Y_i - \hat{\mu}(X_i)) - \beta_\pi \right) \right|$$
$$= \left| (\hat{\beta} - \beta_\pi) - \frac{1}{\alpha} \mathbb{1}\{\hat{\mu}(X_i) \leq \hat{\beta}\} \hat{\beta} + \frac{1}{\alpha} \mathbb{1}\{\hat{\mu}(X_i) \leq \beta_\pi\} \beta_\pi + \frac{1}{\alpha} \mathbb{1}\{\hat{\mu}(X_i) \leq \hat{\beta}\} \beta_\pi - \frac{1}{\alpha} \mathbb{1}\{\hat{\mu}(X_i) \leq \hat{\beta}\} \beta_\pi \right.$$
$$\left. + \frac{1}{\alpha} (\mathbb{1}\{\hat{\mu}(X_i) \leq \hat{\beta}\} - \mathbb{1}\{\hat{\mu}(X_i) \leq \beta_\pi\}) \left( \hat{\mu}(X_i) + \frac{\mathbb{1}\{A_i = \pi(X_i)\}}{\pi_0(\pi(X_i) \mid X_i)} (Y_i - \hat{\mu}(X_i)) - \beta_\pi \right) \right|$$
$$= \left| (\hat{\beta} - \beta_\pi) \left( 1 - \frac{\mathbb{1}\{\hat{\mu}(X_i) \leq \hat{\beta}\}}{\alpha} \right) \right.$$
$$\left. + \frac{1}{\alpha} (\mathbb{1}\{\hat{\mu}(X_i) \leq \hat{\beta}\} - \mathbb{1}\{\hat{\mu}(X_i) \leq \beta_\pi\}) \left( \hat{\mu}(X_i) + \frac{\mathbb{1}\{A_i = \pi(X_i)\}}{\pi_0(\pi(X_i) \mid X_i)} (Y_i - \hat{\mu}(X_i)) - \beta_\pi \right) \right|.$$

By Assumption 2.1, we have that $|\beta_\pi| \leq \bar{y}$. Therefore, Term (2) is bounded by

$$(2) \leq \frac{1}{\alpha} |\hat{\beta} - \beta_\pi| + \frac{4\bar{y}}{\alpha \varepsilon}.$$

Similarly, Term (3) can be written as

$$
(3) = \left| \frac{1}{\alpha} \mathbb{1}\{\hat{\mu}(X_i) \le \beta_\pi\} \left( \hat{\mu}(X_i) + \frac{\mathbb{1}\{A = \pi(X_i)\}}{\pi_0(\pi(X_i) \mid X_i)}(Y - \hat{\mu}(X_i)) - \beta_\pi \right) \right.
$$
$$
\left. - \frac{1}{\alpha} \mathbb{1}\{\mu_\pi(X_i) \le \beta_\pi\} \left( \mu_\pi(X_i) + \frac{\mathbb{1}\{A_i = \pi(X_i)\}}{\pi_0(\pi(X_i) \mid X_i)}(Y_i - \mu_\pi(X_i)) - \beta_\pi \right) \right|
$$
$$
= \left| \frac{1}{\alpha}(\mathbb{1}\{\hat{\mu}(X_i) \le \beta_\pi\} - \mathbb{1}\{\mu_\pi(X_i) \le \beta_\pi\}) \left( \frac{\mathbb{1}\{A_i = \pi(X_i)\}}{\pi_0(\pi(X_i) \mid X_i)}Y_i - \beta_\pi \right) \right.
$$
$$
+ \frac{1}{\alpha} \mathbb{1}\{\hat{\mu}(X_i) \le \beta_\pi\} \left( \left(1 + \frac{\mathbb{1}\{A_i = \pi(X_i)\}}{\pi_0(\pi(X_i) \mid X_i)}\right)\hat{\mu}(X_i) \right)
$$
$$
\left. - \frac{1}{\alpha} \mathbb{1}\{\mu_\pi(X_i) \le \beta_\pi\} \left( \left(1 + \frac{\mathbb{1}\{A_i = \pi(X_i)\}}{\pi_0(\pi(X_i) \mid X_i)}\right)\mu_\pi(X) \right) \right|
$$
$$
= \left| \frac{1}{\alpha}(\mathbb{1}\{\hat{\mu}(X_i) \le \beta_\pi\} - \mathbb{1}\{\mu_\pi(X_i) \le \beta_\pi\}) \left( \frac{\mathbb{1}\{A_i = \pi(X_i)\}}{\pi_0(\pi(X_i) \mid X_i)}Y_i - \beta_\pi \right) \right.
$$
$$
+ \frac{1}{\alpha} \mathbb{1}\{\hat{\mu}(X_i) \le \beta_\pi\} \left( \left(1 + \frac{\mathbb{1}\{A_i = \pi(X_i)\}}{\pi_0(\pi(X_i) \mid X_i)}\right)\hat{\mu}(X_i) \right)
$$
$$
- \frac{1}{\alpha} \mathbb{1}\{\mu_\pi(X_i) \le \beta_\pi\} \left( \left(1 + \frac{\mathbb{1}\{A_i = \pi(X_i)\}}{\pi_0(\pi(X_i) \mid X_i)}\right)\mu_\pi(X_i) \right)
$$
$$
+ \frac{1}{\alpha} \mathbb{1}\{\hat{\mu}(X_i) \le \beta_\pi\} \left( \left(1 + \frac{\mathbb{1}\{A_i = \pi(X_i)\}}{\pi_0(\pi(X_i) \mid X_i)}\right)\mu_\pi(X_i) \right)
$$
$$
\left. - \frac{1}{\alpha} \mathbb{1}\{\hat{\mu}(X_i) \le \beta_\pi\} \left( \left(1 + \frac{\mathbb{1}\{A_i = \pi(X_i)\}}{\pi_0(\pi(X_i) \mid X_i)}\right)\mu_\pi(X_i) \right) \right|.
$$

Rearrange, we have that

$$
(3) \le \left| \frac{1}{\alpha}(\mathbb{1}\{\hat{\mu}(X_i) \le \beta_\pi\} - \mathbb{1}\{\mu_\pi(X_i) \le \beta_\pi\}) \left( \frac{\mathbb{1}\{A_i = \pi(X_i)\}}{\pi_0(\pi(X_i) \mid X_i)}Y_i - \beta_\pi \right) \right|
$$
$$
+ \left| \frac{1}{\alpha} \mathbb{1}\{\hat{\mu}(X_i) \le \beta_\pi\} \left(1 - \frac{\mathbb{1}\{A_i = \pi(X_i)\}}{\pi_0(\pi(X_i) \mid X_i)}\right)(\hat{\mu}(X_i) - \mu_\pi(X_i)) \right|
$$
$$
+ \left| \frac{1}{\alpha}(\mathbb{1}\{\hat{\mu}(X_i) \le \beta_\pi\} - \mathbb{1}\{\mu_\pi(X_i) \le \beta_\pi\}) \left(1 - \frac{\mathbb{1}\{A_i = \pi(X_i)\}}{\pi_0(\pi(X_i) \mid X_i)}\right)\mu_\pi(X_i) \right|
$$
$$
\le \frac{3\bar{y}}{\alpha\varepsilon} + \frac{1}{\alpha\varepsilon}|\hat{\mu}(X_i) - \mu_\pi(X_i)|.
$$

Putting everything together, we have that

$$
|\phi(Z_i; \hat{\pi}_0, \hat{\mu}, \hat{\beta}) - \phi(Z_i; \pi_0, \mu_\pi, \beta_\pi)| \le \frac{2\bar{y}}{\alpha\varepsilon^{3/2}}|\hat{\pi}_0(\pi(X_i) \mid X_i) - \pi_0(\pi(X_i) \mid X_i)| + \frac{1}{\alpha}|\hat{\beta} - \beta_\pi|
$$
$$
+ \frac{1}{\alpha\varepsilon}|\hat{\mu}(X_i) - \mu_\pi(X_i)| + \frac{7\bar{y}}{\alpha\varepsilon}.
$$

$\square$

### F.2 PROOF OF LEMMA E.2

Before we embark on the proof that utilizes a chaining argument Zhou et al. (2023), we present the following definitions that will be needed throughout the analysis.

**Definition F.1** (Rademacher complexity). *Let $\gamma_i$'s be i.i.d. Rademacher random variables $\mathbb{P}(\gamma_i = 1) = \mathbb{P}(\gamma_i = -1) = \frac{1}{2}$.*

1. *The empirical Rademacher complexity $\mathcal{R}_n(\mathcal{F})$ of a function class $\mathcal{F}$ with domain $\mathcal{X}$ is defined as*

$$
\mathcal{R}_n(\mathcal{F} \mid \{X_i \in \mathcal{X}\}_{i=1}^n) = \mathbb{E}_\gamma\left[ \sup_{f \in \mathcal{F}} \frac{1}{n}\left| \sum_{i=1}^n \gamma_i f(X_i) \right| \mid \{X_i \in \mathcal{X}\}_{i=1}^n \right].
$$

2. *The* Rademacher complexity $\mathcal{R}(\mathcal{F})$ *of the function class* $\mathcal{F}$ *is* $\mathbb{E}_X[\mathcal{R}_n(\Pi \mid \{X_i \in \mathcal{X}\}_{i=1}^n)]$.

Before we introduce the chaining technique, we define the Hamming distance $H(\pi_1, \pi_2) = \frac{1}{n}\sum_{j=1}^n \mathbb{1}\{\pi_1 \neq \pi_2\}$, and the Entropy integral $\nu(\Pi)$.

**Definition F.2** ($L_2$ policy distance). *Given a fixed policy class* $\Pi$ *and a set of* $n$ *covariate points* $\{x_1, \cdots, x_n\}$, *we define the following.*

1. *For a function class* $\mathcal{F}_\Pi = \{f(\cdot; \pi) \mid \pi \in \Pi\}$ *such that* $f$ *is a function on* $(Z; \pi)$ *such that* $|f(Z; \pi)| \leq \bar{f}(Z)$, *define* $L_2$ *distance* $D_2(\pi_1, \pi_2; \{Z_1, \cdots, Z_n\})$ *between two policies* $\pi_1, \pi_2$ *with respect to* $\{Z_1, \cdots, Z_n\}$ *is*

$$D_2(\pi_1, \pi_2) = \sqrt{\frac{\sum_{i=1}^n |f(Z_i; \pi_1) - f(Z_i; \pi_2)|^2}{4\sum_{i=1}^n \bar{f}^2(Z_i)}}.$$

2. *The* $\epsilon$-$L_2$ *covering number of the set* $\{x_1, \cdots, x_n\}$ *(denoted as* $N_2(\epsilon, \Pi, \{x_1, \cdots, x_n\})$*) is the smallest number* $N$ *of policies* $\{\pi_1, \cdots, \pi_N\}$ *in* $\Pi$ *such that* $\forall \pi \in \Pi$, *there exists* $\pi_i$ *such that* $D_2(\pi, \pi_i) \leq \epsilon$.

3. *The* $\epsilon$-$L_2$ *covering number of* $\Pi$ *is* $N_{\ell_2}(\epsilon, \Pi) := \sup\{N_2(\gamma, \Pi, \{x_1, \cdots, x_j\}) | j \geq 1, x_1, \cdots, x_j \in \mathcal{X}\}$.

**Policy Chaining.** Conditioned on the data $\{X_1, \cdots, X_n\}$, we define a sequence of refining approximation operators: $A_0, A_1, \cdots, A_J$ where $M = \lceil \log_2 n \rceil$ and each $A_j^\pi : \mathcal{X} \to \mathcal{A}$ is another policy. Define $\underline{J} = \lfloor 1/2 \log_2 n \rfloor$. For each policy $\pi \in \Pi$, we can write it in terms of the approximation policies as

$$\pi(x) = A_0^\pi + \sum_{j=1}^{\underline{J}} (A_j^\pi(x) - A_{j-1}^\pi(x)) + (A_J^\pi(x) - A_{\underline{J}}^\pi(x)) + (\pi(x) - A_J^\pi(x)). \quad (14)$$

We now give an explicit construction of the sequence of approximation operators. Set $\gamma_j = \frac{1}{2^j}$ and let $S_0, S_1, \cdots, S_J$ be a sequence of policy classes (understood to be subclasses of $\Pi$) such that $S_j$ could $\gamma_j$-cover $\Pi$ under the inner product distance:

$$\forall \pi \in \Pi, \exists \pi' \in S_j, D_2(\pi, \pi') \leq \gamma_j.$$

By Definition F.2, we can choose the $m$-th policy class $S_m$ such that $|S_j| = N_2(2^{-j}, \Pi, \{X_1, \cdots, X_n\})$. Note that in particular $|S_0| = 1$, since any single policy is enough to 1-cover all policies in $\Pi$.

Next, we use the following backward selection scheme to define $A_j$'s. For each $\pi \in \Pi$, define

$$A_J^\pi = \arg\min_{\pi' \in S_J} D_2(\pi, \pi').$$

Further, for each $0 \leq j < J$ and each $\pi \in \Pi$, inductively define

$$A_j^\pi = \arg\min_{\pi' \in S_j} D_2(A_{j+1}^\pi, \pi').$$

Appendix G presents a few helper results that would facilitate the following theorem, which is needed for the proof of Lemma E.2.

**Theorem F.3.** *Suppose that* $\mathcal{F}_\Pi := \{f(\cdot; \pi) \mid \pi \in \Pi\}$ *is a function class of* $f(\cdot; \pi)$ *that takes* $Z$ *as input. Given a set of dataset* $\mathcal{D} = \{Z_i = (X_i, A_i, Y_i)\}_{i=1}^n$, *suppose that* $|f(Z_i; \pi(X_i))|_\infty \leq \bar{f}(Z_i)$. *Then the Rademacher complexity*

$$\mathcal{R}_n(\mathcal{F}_\Pi) \leq \frac{8\sqrt{\sum_{i=1}^n \bar{f}^2(Z_i)}}{n}(\kappa(\Pi) + 7) + \frac{6\sqrt{\sum_{i=1}^n \bar{f}^2(Z_i)}}{n} + o\left(\frac{1}{\sqrt{n}}\right).$$

*Proof.* We will investigate the Rademacher complexity of the function class $\mathcal{F}_\Pi := \{f(\cdot, \pi) \mid \pi \in \Pi\}$. Each policy $\pi \in \Pi$ can be written in terms of the approximation policies as in equation 14.

Accordingly, we can expand the Rademacher complexity

$$\mathcal{R}_n(\mathcal{F}_\Pi) = \mathbb{E}_\epsilon\left[\sup_{\pi\in\Pi}\frac{1}{n}\left|\sum_{i=1}^n \epsilon_i f(Z_i;\pi)\right|\right]$$

$$=\mathbb{E}_\epsilon\left[\sup_{\pi\in\Pi}\frac{1}{n}\left|\sum_{i=1}^n \epsilon_i\left(f(Z_i;A_0^\pi) + \sum_{j=1}^J \big(f(Z_i;A_j^\pi) - f(Z_i;A_{j-1}^\pi)\big) + \big(f(Z_i;\pi) - f(Z_i;A_J^\pi)\big)\right)\right|\right]$$

$$\leq\mathbb{E}_\epsilon\left[\sup_{\pi\in\Pi}\frac{1}{n}\left|\sum_{i=1}^n \epsilon_i f(Z_i;A_0^\pi)\right|\right] + \mathbb{E}_\epsilon\left[\sup_{\pi\in\Pi}\frac{1}{n}\left|\sum_{i=1}^n \epsilon_i\big(f(Z_i;\pi) - f(Z_i;A_J^\pi)\big)\right|\right]$$

$$+\mathbb{E}_\epsilon\left[\sup_{\pi\in\Pi}\frac{1}{n}\left|\sum_{i=1}^n \epsilon_i\big(\sum_{j=1}^J f(Z_i;A_j^\pi) - f(Z_i;A_{j-1}^\pi)\big)\right|\right].$$

We first note that the first term

$$\mathbb{E}_\epsilon\left[\sup_{\pi\in\Pi}\frac{1}{n}\left|\sum_{i=1}^n \epsilon_i f(Z_i;A_0^\pi)\right|\right] = \mathbb{E}_\epsilon\left[\frac{1}{n}\left|\sum_{i=1}^n \epsilon_i f(Z_i;\bar\pi)\right|\right],$$

as $A_0^\pi$ maps all $\pi\in\Pi$ to a singular policy $\bar\pi$. Since $|\epsilon_i f(Z_i;\bar\pi)| \leq \bar{f}(Z_i)$, by Azuma-Hoeffding's lemma, we have that

$$\mathbb{P}\left(\frac{1}{n}\left|\sum_{i=1}^n \epsilon_i f(Z_i;\bar\pi)\right| \geq t\right) \leq 2\exp\left(-\frac{n^2 t^2}{2\sum_{i=1}^n \bar{f}^2(Z_i)}\right).$$

Therefore, the expectation

$$\mathbb{E}_\epsilon\left[\frac{1}{n}\left|\sum_{i=1}^n \epsilon_i f(Z_i;\bar\pi)\right|\right] = \int_0^\infty \mathbb{P}_\epsilon\left(\frac{1}{n}\left|\sum_{i=1}^n \epsilon_i f(Z_i;\bar\pi)\right| \geq t\right) dt \leq \int_0^\infty 2\exp\left(-\frac{n^2 t^2}{2\sum_{i=1}^n \bar{f}^2(Z_i)}\right) dt$$

$$=\frac{6\sqrt{\sum_{i=1}^n \bar{f}^2(Z_i)}}{n}.$$

We will bound the other terms separately in the following steps.

**The Negligible Regime.** In this step, we establish two claims to show that $\pi - A_M(\pi)$ is in the negligible regimes. For any $\pi\in\Pi$, by the Cauchy-Schwarz inequality,

$$\sup_{\pi\in\Pi}\left|\frac{1}{n}\sum_{i=1}^n \epsilon_i|f(Z_i;\pi) - f(Z_i;A_J^\pi)|\right| \leq \frac{1}{n}\sqrt{n\sum_{i=1}^n \big(f(Z_i;\pi) - f(Z_i;A_J^\pi)\big)^2}$$

$$=\frac{2\sqrt{\sum_{i=1}^n \bar{f}^2(Z_i)}}{\sqrt{n}} D_2(\pi, A_J^\pi; \{Z_1,\cdots,Z_n\})$$

$$\leq\frac{2\sqrt{\sum_{i=1}^n \bar{f}^2(Z_i)}}{\sqrt{n}}2^{-J} \leq \frac{2\sqrt{\sum_{i=1}^n \bar{f}^2(Z_i)}}{n^{\frac{3}{2}}},$$

where the second-to-last step is due to the fact that the policy $A_M^\pi$ is $2^{-M}$-close to $\pi$ and the last step is due to the definition of $M$. Therefore, we conclude that the term

$$\mathbb{E}_\epsilon\left[\sup_{\pi\in\Pi}\frac{1}{n}\left|\sum_{i=1}^n \epsilon_i\big(f(Z_i;\pi) - f(Z_i;A_J^\pi)\big)\right|\right] \leq \frac{2\sqrt{\sum_{i=1}^n \bar{f}^2(Z_i)}}{n^{\frac{3}{2}}},$$

and is in the negligible regime.

**The Effective Regime.** By the previous results, we have that

$$\mathcal{R}_n(\mathcal{F}_\Pi) = \mathbb{E}_\epsilon\left[\sup_{\pi\in\Pi}\frac{1}{n}\left|\sum_{i=1}^n \epsilon_i f(Z_i;\pi)\right|\right]$$

$$\leq\mathbb{E}_\epsilon\left[\sup_{\pi\in\Pi}\frac{1}{n}\left|\sum_{i=1}^n \epsilon_i\big(\sum_{j=1}^J f(Z_i;A_j^\pi) - f(Z_i;A_{j-1}^\pi)\big)\right|\right] + o\left(\frac{1}{\sqrt{n}}\right).$$

From now on, for easier notation, we denote $\Lambda = 2\sqrt{\sum_{i=1}^n \bar{f}^2(Z_i)}$. We will now concentrate on the expectation in the above inequality. Let $P_m$ denote the projection of a policy to $S_j$, for $A_{j-1}^\pi = P_{j-1}(A_j^\pi)$ for all $j \in [J]$. Note that once $A_j^\pi$ is determined, the policy $A_{j-1}^\pi$ is also determined. For any $t > 0$,

$$\mathbb{P}_\epsilon\left(\sup_{\pi \in \Pi}\left|\frac{1}{n}\sum_{i=1}^n \epsilon_i\big(f(Z_i; A_j^\pi) - f(Z_i; A_{j-1}^\pi)\big)\right| \geq t\right)$$

$$\leq \sum_{\pi' \in S_j} \mathbb{P}_\epsilon\left(\left|\frac{1}{n}\sum_{i=1}^n \epsilon_i\big(f(Z_i; \pi') - f(Z_i; P_{j-1}(\pi'))\big)\right| \geq t\right)$$

$$\leq \sum_{\pi' \in S_j} 2 \cdot \exp\left(-\frac{2n^2 t^2}{\sum_{i=1}^n (f(Z_i; \pi') - f(Z_i; P_{j-1}(\pi')))^2}\right)$$

$$= \sum_{\pi' \in S_j} 2 \cdot \exp\left(-\frac{2nt^2}{\lambda^2 D_2^2(\pi', P_{j-1}(\pi'); Z)}\right)$$

$$\leq 2N_2(2^{-j}, \Pi; \mathcal{D}) \cdot \exp\left(-\frac{n^2 t^2}{\Lambda^2 D_2(\pi', P_{j-1}(\pi'); Z)^2}\right).$$

For any $j = 1, \cdots, J$ and $p \in \mathbb{N}$, let $t_{j,p} = \frac{\Lambda}{n2^{j-1/2}}\sqrt{\log(2^{p+1}j^2 \cdot N_2(2^{-j}, \Pi; \mathcal{D}))}$. Then for a fixed $p$, with a union bound over $j = 1, \cdots, J$, we have that

$$\mathbb{P}_\epsilon\left(\sup_{\pi \in \Pi}\left|\sum_{j=1}^J \frac{1}{n}\sum_{i=1}^n \epsilon_i(f(Z_i; A_j^\pi) - f(Z_i; A_{j-1}^\pi))\right| \geq \sum_{j=1}^J t_{j,p}\right)$$

$$\leq \sum_{j=1}^J \mathbb{P}_\epsilon\left(\sup_{\pi \in \Pi}\left|\sum_{j=1}^J \frac{1}{n}\sum_{i=1}^n \epsilon_i(f(Z_i; A_j^\pi) - f(Z_i; A_{j-1}^\pi))\right| \geq t_{j,p}\right) \leq \sum_{j=1}^J \frac{1}{j^2 2^p} \leq \frac{1}{2^{p-1}}.$$

Using helper Proposition G.1, for any $j \in \mathbb{N}$,

$$\sum_{j=1}^J t_{j,p} = \sum_{j=1}^J \frac{\Lambda}{2^{j-1/2}n}\sqrt{\log(2^{p+1}j^2 \cdot N_2(2^{-j}, \Pi; \mathcal{D}))}$$

$$\leq \sum_{j=1}^J \frac{\Lambda}{2^{j-1/2}n}\sqrt{\log(N_2(2^{-j}, \Pi; \mathcal{D})) + (p+1)\log 2 + 2\log j}$$

$$\leq \frac{2\Lambda}{n}\sum_{j=1}^J 2^{-j}\big(\sqrt{\log(N_2(2^{-j}, \Pi; \mathcal{D}))} + \sqrt{(p+1)\log 2} + \sqrt{2\log j}\big)$$

$$\leq \frac{4\Lambda}{n}(\kappa(\Pi) + \sqrt{p+1} + 1) =: t_p,$$

where the first inequality is uses the fact that $\sqrt{a+b+c} \leq \sqrt{a} + \sqrt{b} + \sqrt{c}$ for $a, b, c \geq 0$; and the last inequality is due to the definition of $\kappa(\Pi)$. Then

$$\mathbb{E}_\epsilon\left[\sup_{\pi \in \Pi}\frac{1}{n}\left|\sum_{i=1}^n \epsilon_i\big(\sum_{j=1}^J f(Z_i; A_j^\pi) - f(Z_i; A_{j-1}^\pi)\big)\big)\right|\right]$$

$$= \int_0^\infty \mathbb{P}_\epsilon\left(\sup_{\pi \in \Pi}\left|\sum_{j=1}^J \frac{1}{n}\sum_{i=1}^n \epsilon_i\big(f(Z_i; A_j^\pi) - f(Z_i; A_{j-1}^\pi)\big)\right| > t\right) dt$$

$$\leq t_1 + \sum_{p=1}^\infty 2^{-p+1}(u_{p+1} - u_p) = \frac{4\Lambda}{n}\left(\kappa(\Pi) + \sqrt{2} + 1 + \sum_{p=1}^\infty 2^{-p+1}(\sqrt{p+2} - \sqrt{p+1})\right)$$

$$\leq \frac{4\Lambda}{n}(\kappa(\Pi) + 7).$$

Putting everything together, we have that

$$\mathcal{R}_n(\mathcal{F}_\Pi) \leq \frac{8\sqrt{\sum_{i=1}^n \bar{f}^2(Z_i)}}{n}(\kappa(\Pi) + 7) + \frac{6\sqrt{\sum_{i=1}^n \bar{f}^2(Z_i)}}{n} + o\left(\frac{1}{\sqrt{n}}\right).$$

□

Define the oracle policy CVaR estimator with the true $\pi_0, \{\mu_a, a \in \mathcal{A}\}$, and the oracle policy VaR $\tilde{\beta}_\pi$ derived from equation 12:

$$\tilde{\mathcal{V}}_\alpha(\pi) := \frac{1}{n} \sum_{i \in \mathcal{D}} \phi(\pi, Z_i; \pi_0, \{\mu_a\}_{a \in \{0,1\}}, \tilde{\beta}_\pi) =: \frac{1}{n} \sum_{i \in \mathcal{D}} \tilde{\phi}(\pi, Z_i),$$

where $\mu_\pi(x) = \mu_{\pi(x)}(x)$ is constructed from $\{\mu_a, a \in \mathcal{A}\}$. Define $\tilde{\mathcal{F}}_\Pi := \{\tilde{\phi}(\cdot; \pi) \mid \pi \in \Pi\}$. The following corollary bounds the Rademacher complexity of $\tilde{\mathcal{F}}_\Pi$.

**Corollary F.4.** *Under Assumption 2.1 and 3.3,*

$$\mathcal{R}_n(\tilde{\mathcal{F}}_\Pi) \leq \mathcal{R}_n(\mathcal{F}_\Pi) \leq \frac{8\bar{y}}{\sqrt{n}}(\kappa(\Pi) + 7) + \frac{6\bar{y}}{\sqrt{n}} + o\left(\frac{1}{\sqrt{n}}\right).$$

*Proof.* We apply Theorem F.3 with function class $\tilde{\mathcal{F}}_\Pi$, in which each function $\|\tilde{\phi}\|_{L_\infty} \leq \bar{y}$. □

We are now ready to prove Lemma E.2.

*Proof of Lemma E.2.* We first note that for any $\pi \in \Pi$, the expectation of the oracle policy value $\tilde{\mathcal{V}}_\alpha(\pi)$,

$$\mathbb{E}[\tilde{\mathcal{V}}_\alpha(\pi)]$$

$$= \mathbb{E}\left[\frac{1}{n} \sum_{i=1}^n \phi(\pi, Z_i; \pi_0, \mu_\pi, \beta_\pi)\right]$$

$$= \mathbb{E}\left[\frac{1}{n} \sum_{i=1}^n \left(\beta_\pi + \frac{1}{\alpha}\mathbb{1}\{\mu_\pi(X_i) \leq \beta_\pi\}\left(\mu_\pi(X_i) + \frac{\mathbb{1}\{A_i = \pi(X_i)\}}{\pi_0(\pi(X_i) \mid X_i)}(Y_i - \mu_\pi(X_i)) - \beta_\pi\right)\right)\right]$$

$$= \mathbb{E}\left[\mathbb{E}\left[\beta_\pi + \frac{1}{\alpha}\mathbb{1}\{\mu_\pi(X_i) \leq \beta_\pi\}\left(\mu_\pi(X_i) + \frac{\mathbb{1}\{A_i = \pi(X_i)\}}{\pi_0(\pi(X_i) \mid X_i)}(Y_i - \mu_\pi(X_i)) - \beta_\pi\right) \mid X = X_i\right]\right]$$

$$= \mathbb{E}\left[\beta_\pi + \frac{1}{\alpha}\mathbb{1}\{\mu_\pi(X) \leq \beta_\pi\}(Y(\pi(X)) - \beta_\pi)\right] = \beta_\pi + \frac{1}{\alpha}\mathbb{E}[\mathbb{1}\{\mu_\pi(X) \leq \beta_\pi\}(Y(\pi(X)) - \beta_\pi)]$$

$$= \mathcal{V}_\alpha(\pi).$$

To see the last equality, we note that, for the underlying true $\beta_\pi$ of a policy $\pi \in \Pi$,

$$\beta_\pi + \frac{1}{\alpha}\mathbb{E}[\mathbb{1}\{\mu_\pi(X) \leq \beta_\pi\}(Y(\pi(X)) - \beta_\pi)]$$

$$= \beta_\pi + \frac{1}{\alpha}\mathbb{E}[\mathbb{1}\{\mu_\pi(X) \leq \beta_\pi\}Y(\pi(X))] - \frac{1}{\alpha}\mathbb{P}(\mu_\pi(X) \leq \beta_\pi)\beta_\pi$$

$$= \beta_\pi + \frac{1}{\alpha}\mathbb{E}[\mathbb{1}\{\mu_\pi(X) \leq \beta_\pi\}Y(\pi(X))] - \frac{\alpha}{\alpha}\beta_\pi$$

$$= \frac{1}{\alpha}\mathbb{E}[\mathbb{1}\{\mu_\pi(X) \leq \beta_\pi\}Y(\pi(X))]$$

The policy value $\mathcal{V}_\alpha$ is defined as the CVaR of policy $\pi$, and the dual formulation Rockafellar et al. (2000) of which is

$$\mathcal{V}_\alpha(\pi) = \inf_{0 \leq V \leq 1, \mathbb{E}[V]=1} \frac{1}{\alpha}\mathbb{E}[VY(\pi(X))],$$

where we define $V := \mathbb{1}\{\mu(X) \le \beta\}$ for some $\mu, \beta$. The above infimum is achieved by the true $\mu_\pi$ and $\beta_\pi$.

Recall that $\|\phi(\pi, Z_i; \pi_0, \mu_\pi, \beta_\pi)\|_{L_\infty} \le \bar{y}$. We apply Theorem 4.10 in Wainwright (2019) with results as Corollary F.4,

$$\sup_{\pi \in \Pi} |\mathcal{V}_\alpha(\pi) - \tilde{\mathcal{V}}_\alpha(\pi)| = \sup_{\pi \in \Pi} \left| \frac{1}{n} \sum_{i=1}^n \phi(\pi, Z_i; \pi_0, \mu_\pi, \beta_\pi) - \mathbb{E}[\phi(\pi, Z_i; \pi_0, \mu_\pi, \beta_\pi)] \right|$$

$$\le 2R_n(\tilde{\mathcal{F}}_\Pi) + \bar{y}\sqrt{\frac{2}{n}}$$

$$\le \frac{16\bar{y}}{\sqrt{n}}(\kappa(\Pi) + 7) + \frac{(12 + \sqrt{2})\bar{y}}{\sqrt{n}} + o\left(\frac{1}{\sqrt{n}}\right),$$

with probability at least $1 - \Delta$. $\qquad\square$

### F.3 PROOF OF COROLLARY E.3

*Proof.* For any policy $\pi \in \Pi$ and any fold $k \in [K]$, we decompose:

$$\hat{\mathcal{V}}_\alpha^{(k)}(\pi) - \mathcal{V}_\alpha(\pi) = \frac{1}{|\mathcal{D}^{(k)}|} \sum_{i \in \mathcal{D}^{(k)}} \phi(\pi, Z_i; \hat{\pi}_0^{(k)}, \hat{\mu}_\pi^{(k)}, \hat{\beta}_\pi^{(k)}) - \phi(\pi, Z_i; \pi_0, \mu_\pi, \beta_\pi)$$

$$= \frac{1}{|\mathcal{D}^{(k)}|} \sum_{i \in \mathcal{D}^{(k)}} \phi(\pi, Z_i; \hat{\pi}_0^{(k)}, \hat{\mu}_\pi^{(k)}, \hat{\beta}_\pi^{(k)}) - \mathbb{E}[\phi(\pi, Z; \hat{\pi}_0^{(k)}, \hat{\mu}_\pi^{(k)}, \hat{\beta}_\pi^{(k)}) \mid \bar{\mathcal{D}}^{(k)}] - \phi(\pi, Z_i; \pi_0, \mu_\pi, \beta_\pi)$$

$$+ \mathbb{E}[\phi(\pi, Z; \pi_0, \mu_\pi, \beta_\pi) \mid \bar{\mathcal{D}}^{(k)}] + \mathbb{E}[\phi(\pi, Z; \hat{\pi}_0^{(k)}, \hat{\mu}_\pi^{(k)}, \hat{\beta}_\pi^{(k)}) \mid \bar{\mathcal{D}}^{(k)}] - \mathbb{E}[\phi(\pi, Z; \pi_0, \mu_\pi, \beta_\pi) \mid \bar{\mathcal{D}}^{(k)}]$$

$$= \mathbb{E}[\phi(\pi, Z; \hat{\pi}_0^{(k)}, \hat{\mu}_\pi^{(k)}, \hat{\beta}_\pi^{(k)}) \mid \bar{\mathcal{D}}^{(k)}] - \mathbb{E}[\phi(\pi, Z; \pi_0, \mu_\pi, \beta_\pi) \mid \bar{\mathcal{D}}^{(k)}]$$

$$+ \frac{1}{|\mathcal{D}^{(k)}|} \sum_{i \in \mathcal{D}^{(k)}} \bigg( \phi(\pi, Z_i; \hat{\pi}_0^{(k)}, \hat{\mu}_\pi^{(k)}, \hat{\beta}_\pi^{(k)}) - \phi(\pi, Z_i; \pi_0, \mu_\pi, \beta_\pi)$$

$$- \big( \mathbb{E}[\phi(\pi, Z; \hat{\pi}_0^{(k)}, \hat{\mu}_\pi^{(k)}, \hat{\beta}_\pi^{(k)}) \mid \bar{\mathcal{D}}^{(k)}] - \mathbb{E}[\phi(\pi, Z; \pi_0, \mu_\pi, \beta_\pi) \mid \bar{\mathcal{D}}^{(k)}] \big) \bigg) =: (I) + (II).$$

We will bound the two terms separately, with fixed $\pi \in \Pi, k \in [K]$.

Let $d_1(\pi, Z_i) := \phi(Z_i; \hat{\pi}_0^{(k)}, \hat{\mu}_\pi^{(k)}, \hat{\beta}_\pi^{(k)}) - \phi(Z_i; \pi_0, \mu_\pi, \beta_\pi)$. By Lemma E.1,

$$\sup_{\pi \in \Pi} |(I)| = \sup_{\pi \in \Pi} |\mathbb{E}[d_1(\pi, Z) \mid \bar{\mathcal{D}}^{(k)}]|$$

$$\le \sup_{\pi \in \Pi} \bigg| \frac{2\bar{y}}{\alpha\varepsilon} \|\hat{\pi}_0^{(k)} - \pi_0\|_{L_2(P)} \|\hat{\mu}_\pi^{(k)} - \mu_\pi\|_{L_2(P)}$$

$$+ \frac{1}{\alpha}(F'_{\mu_\pi(X)}(F^{-1}_{\mu_\pi(X)}(\alpha)) + 1)(\|\hat{\mu}_\pi^{(k)} - \mu_\pi\|_{L_\infty} + |\beta_\pi - \hat{\beta}_\pi^{(k)}|)^2$$

$$+ \frac{1}{2\alpha}(F'_{\mu_\pi(X)}(F^{-1}_{\mu_\pi(X)}(\alpha)) + 1)|\hat{\beta}_\pi^{(k)} - \beta_\pi|^2 \bigg|$$

$$\le \frac{2\bar{y}}{\alpha\varepsilon} \big( \max_{a \in \mathcal{A}} \|\hat{\pi}_0^{(k)}(a \mid X) - \pi_0(a \mid X)\|_{L_2(P)} \big) \big( \max_{a \in \mathcal{A}} \|\hat{\mu}_a^{(k)} - \mu_a\|_{L_2(P)} \big)$$

$$+ \frac{2\bar{F}_\alpha}{\alpha} \bigg( \max_{a \in \mathcal{A}} \|\hat{\mu}_a^{(k)} - \mu_a\|_{L_\infty} + \max_{\pi \in \Pi} |\beta_\pi - \hat{\beta}_\pi^{(k)}| \bigg)^2 + \frac{\bar{F}_\alpha}{\alpha} \max_{\pi \in \Pi} |\hat{\beta}_\pi^{(k)} - \beta_\pi|^2.$$

Applying Lemma 3.4, there exists some $N_\beta \in \mathbb{Z}_+$ such that when $n > n_1$, with probability at least $1 - \Delta$,

$$\max_{\pi \in \Pi} |\hat{\beta}_\pi^{(k)} - \beta_\pi| < \max_{a \in \mathcal{A}} \|\hat{\mu}_a^{(k)} - \mu_a\|_{L_\infty},$$

which means

$$\sup_{\pi \in \Pi} |(I)| \le \frac{2\bar{y}}{\alpha\varepsilon} \big( \max_{a \in \mathcal{A}} \|\hat{\pi}_0^{(k)}(a \mid X) - \pi_0(a \mid X)\|_{L_2(P)} \big) \big( \max_{a \in \mathcal{A}} \|\hat{\mu}_a^{(k)} - \mu_a\|_{L_2(P)} \big)$$

$$+ \frac{8\bar{F}_\alpha}{\alpha} \max_{a \in \mathcal{A}} \|\hat{\mu}_a^{(k)} - \mu_a\|_{L_\infty}^2 + \frac{\bar{F}_\alpha}{\alpha} \max_{a \in \mathcal{A}} \|\hat{\mu}_a^{(k)} - \mu_a\|_{L_\infty}^2.$$

On the event of Lemma 3.4, by Assumption 3.2, there exists some $n_1 \in \mathbb{Z}_+$ such that when $n \geq n_1$ with probability at least $1 - \Delta$,

$$\sup_{\pi \in \Pi} |(I)| \leq \frac{2\bar{y} + 9\bar{F}_\alpha}{\alpha \varepsilon \sqrt{n}}.$$

In summary, there exists some $N_1 = \max\{n_1, N_\beta\}$ such that when $n \geq N_1$, with probability at least $1 - 2K\Delta$, the above inequality holds.

We now turn to Term $(II)$. Let $d_2(\pi, Z_i) := d_1(\pi, Z_i) - \mathbb{E}[d_1(\pi, Z) \mid \bar{\mathcal{D}}^{(k)}]$. Note that Term $(II)$ is zero-mean:

$$\mathbb{E}[(II))] = \mathbb{E}\big[(\hat{\mathbb{E}}_k - \mathbb{E}_{|\bar{k}})[d(\pi, Z)]\big] = \mathbb{E}\bigg[\frac{1}{|\mathcal{D}^{(k)}|} \sum_{i \in \mathcal{D}^{(k)}} d_2(\pi, Z_i)\bigg] = \mathbb{E}[d_1(\pi, Z)] - \mathbb{E}[d_1(\pi, Z)] = 0.$$

By Lemma E.1,

$$\big|d_1(\pi, Z_i)\big| \leq \frac{4\bar{y}}{\alpha \varepsilon^{3/2}} |\hat{\pi}_0^{(k)}(\pi(X_i) \mid X_i) - \pi_0(\pi(X_i) \mid X_i)| + \frac{2}{\alpha \varepsilon} |\hat{\mu}_\pi^{(k)}(X_i) - \mu_\pi(X_i)| + \frac{1}{\alpha} |\hat{\beta}_\pi^{(k)} - \beta_\pi| + \frac{14\bar{y}}{\alpha \varepsilon}$$

$$\leq \frac{4\bar{y}}{\alpha \varepsilon^{3/2}} \max_{a \in \mathcal{A}} |\hat{\pi}_0^{(k)}(a \mid X_i) - \pi_0(a \mid X_i)| + \frac{2}{\alpha \varepsilon} \max_{a \in \mathcal{A}} |\hat{\mu}_a^{(k)}(X_i) - \mu_a(X_i)| + \frac{1}{\alpha} |\hat{\beta}_\pi^{(k)} - \beta_\pi| + \frac{14\bar{y}}{\alpha \varepsilon}.$$

Applying Lemma 3.4, there exists some $C_1 > 0, N_\beta \in \mathbb{Z}_+$ such that when $n > N_\beta$, with probability at least $1 - \Delta$,

$$\big|d_1(\pi, Z_i)\big| \leq \frac{4\bar{y}}{\alpha \varepsilon^{3/2}} \max_{a \in \mathcal{A}} |\hat{\pi}_0^{(k)}(a \mid X_i) - \pi_0(a \mid X_i)| + \frac{2}{\alpha \varepsilon} \max_{a \in \mathcal{A}} |\hat{\mu}_a^{(k)}(X_i) - \mu_a(X_i)|$$

$$+ \frac{1}{\alpha}\big(n^{-\frac{1}{2}} \vee \max_{a \in \mathcal{A}} \|\hat{\mu}_a^{(k)}(X_i) - \mu_a(X_i)\|_{L_2(P)} + n^{-\frac{1}{4}}\big) + \frac{14\bar{y}}{\alpha \varepsilon}$$

$$\leq \frac{14\bar{y}}{\alpha \varepsilon^{3/2}} \max_{a \in \mathcal{A}} \big(|\hat{\pi}_0^{(k)}(a \mid X_i) - \pi_0(a \mid X_i)| + |\hat{\mu}_a^{(k)}(X_i) - \mu_a(X_i)| + 1\big)$$

$$+ \frac{1}{\alpha}\big(n^{-\frac{1}{2}} \vee \max_{a \in \mathcal{A}} \|\hat{\mu}_a^{(k)}(X_i) - \mu_a(X_i)\|_{L_2(P)}\big) =: \bar{d}_1(Z_i).$$

Consequently,

$$|d_2(\pi, Z_i)| = |d_1(\pi, Z_i) - \mathbb{E}[d_1(\pi, Z_i)]| \leq 2\bar{d}_1(X_i) := \bar{d}_2(X_i).$$

We now apply the bounded difference inequality in (Wainwright, 2019, Corollary 2.21) conditional on $X = \{X_i\}_{i \in [n]}$,

$$\mathbb{P}\bigg(\sup_{\pi \in \Pi}\bigg|\frac{1}{|\mathcal{D}^{(k)}|} \sum_{i \in \mathcal{D}^{(k)}} d_2(\pi, Z_i)\bigg| - \mathbb{E}\bigg[\sup_{\pi \in \Pi}\bigg|\frac{1}{|\mathcal{D}^{(k)}|} \sum_{i \in \mathcal{D}^{(k)}} d_2(\pi, Z_i)\bigg| \mid X\bigg] \geq t \bigg| X\bigg)$$

$$\leq \exp\bigg(-\frac{2|\mathcal{D}^{(k)}|^2 t^2}{\sum_{i \in \mathcal{D}^{(k)}} \bar{d}_2^2(Z_i)}\bigg).$$

Setting $t = \frac{\sqrt{\sum_{i \in \mathcal{D}^{(k)}} \bar{d}_2^2(Z_i) \log(1/\Delta)}}{|\mathcal{D}^{(k)}|}$, then with probability at least $1 - \Delta$,

$$\sup_{\pi \in \Pi}\bigg|\frac{1}{|\mathcal{D}^{(k)}|} \sum_{i \in \mathcal{D}^{(k)}} d_2(\pi, Z_i)\bigg| \leq \mathbb{E}\bigg[\sup_{\pi \in \Pi}\bigg|\frac{1}{|\mathcal{D}^{(k)}|} \sum_{i \in \mathcal{D}^{(k)}} d_2(\pi, Z_i)\bigg| \mid X\bigg] + \frac{\sqrt{\sum_{i \in \mathcal{D}^{(k)}} \bar{d}_2^2(Z_i) \log(1/\Delta)}}{|\mathcal{D}^{(k)}|}.$$

Next, we turn to the expectation in the above inequality.

$$\mathbb{E}\bigg[\sup_{\pi \in \Pi}\bigg|\frac{1}{|\mathcal{D}^{(k)}|} \sum_{i \in \mathcal{D}^{(k)}} d_2(\pi, Z_i)\bigg| \mid X\bigg] \leq R_n(\mathcal{F}_\Pi(d_2)),$$

where we denote $\mathcal{F}_\Pi(d_2) = \{d_2(\pi, \cdot) \mid \pi \in \Pi\}$, in which $|d_2(\pi, Z_i)| \leq \bar{d}_2(Z_i)$. Applying Theorem F.3, we have that

$$\mathcal{R}_n(\mathcal{F}_\Pi(d_2)) \leq \mathcal{R}_n(\mathcal{F}_\Pi) \leq \frac{8\sqrt{\sum_{i=1}^n \bar{d}_2^2(Z_i)}}{n}(\kappa(\Pi) + 7) + \frac{6\sqrt{\sum_{i=1}^n \bar{d}_2^2(Z_i)}}{n} + o\bigg(\frac{1}{\sqrt{n}}\bigg).$$

Consequently, with probability $1 - \Delta$,

$$\sup_{\pi \in \Pi} \frac{1}{|\mathcal{D}^{(k)}|} \left| \sum_{i \in \mathcal{D}^{(k)}} d_2(\pi, Z_i) \right| \leq \frac{\sqrt{\sum_{i=1}^{n} \bar{d}_2^2(Z_i)}}{|\mathcal{D}^{(k)}|} (8\kappa(\Pi) + 62 + \sqrt{\log(1/\Delta)}).$$

Now let $e(a, X_i) := (\hat{\pi}_0^{(k)}(a \mid X_i) - \pi_0(a \mid X_i))^2 + (\hat{\mu}_a^{(k)}(X_i) - \mu_a(X_i))^2$. Since $e(a, X_i) \leq 1 + 4\bar{y}^2$, applying Hoeffding's inequality gives that

$$\mathbb{P}\left( \frac{1}{|\mathcal{D}^{(k)}|} \sum_{i \in \mathcal{D}^{(k)}} \max_{a \in \mathcal{A}} e(a, X_i) - \sum_{a \in \mathcal{A}} \mathbb{E}[e(a, X)] \geq t \right)$$

$$\leq \mathbb{P}\left( \frac{1}{|\mathcal{D}^{(k)}|} \sum_{i \in \mathcal{D}^{(k)}} \sum_{a \in \mathcal{A}} e(a, X_i) - \sum_{a \in \mathcal{A}} \mathbb{E}[e(a, X)] \geq t \right)$$

$$\leq \sum_{a \in \mathcal{A}} \mathbb{P}\left( \frac{1}{|\mathcal{D}^{(k)}|} \sum_{i \in \mathcal{D}^{(k)}} e(a, X_i) - \mathbb{E}[e(a, X)] \geq t \right) \leq M(1 + 4\bar{y}^2) \exp\left( -2|\mathcal{D}^{(k)}| t^2 \right),$$

recalling that $|\mathcal{A}| = M$. Taking a union bound, with probability at least $1 - 2\Delta$, we have that

$$\sup_{\pi \in \Pi} |(II)| \leq \frac{28\bar{y}}{\alpha\varepsilon\sqrt{|\mathcal{D}^{(k)}|}} (8\kappa(\Pi) + 62 + \sqrt{\log(1/\Delta)})$$

$$\times \left( \sum_{a \in \mathcal{A}} \|\hat{\pi}_0^{(k)} - \pi_0\|_{L_2(P)} + \|\hat{\mu}_a^{(k)} - \mu_a\|_{L_2(P)} + 1 + \sqrt[4]{\frac{\log(M(1 + 4\bar{y}^2)/\Delta)}{2|\mathcal{D}^{(k)}|}} \right)$$

$$+ \frac{28\bar{y}}{\alpha\varepsilon\sqrt{|\mathcal{D}^{(k)}|}} (8\kappa(\Pi) + 62 + \sqrt{\log(1/\Delta)}) \times \left( n^{-\frac{1}{2}} \vee \max_{a \in \mathcal{A}} \|\hat{\mu}_a^{(k)}(X_i) - \mu_a(X_i)\|_{L_2(P)} \right).$$

By Assumption 3.2 $\sum_{a \in \mathcal{A}} \|\hat{\pi}_0^{(k)} - \pi_0\|_{L_2(P)} + \|\hat{\mu}_a^{(k)-\mu_a}\|_{L_2(P)} = o_p(1)$. Then there exists some $n_2 \in \mathbb{Z}_+$ such that when $n \geq n_2$, with probability at least $1 - 4K\Delta$,

$$\sup_{\pi \in \Pi} |(II)| \leq \frac{28\bar{y}}{\alpha\varepsilon\sqrt{|\mathcal{D}^{(k)}|}} (8\kappa(\Pi) + 62 + \sqrt{\log(1/\Delta)}) + o\left( \frac{1}{\sqrt{n}} \right).$$

Putting everything together, and setting $\Delta' = 6K\Delta$, with probability at least $1 - \Delta'$,

$$\sup_{\pi \in \Pi} |\hat{\mathcal{V}}_\alpha(\pi) - \mathcal{V}_\alpha(\pi)| \leq \frac{28\bar{y}}{\alpha\varepsilon\sqrt{n}} (8\kappa(\Pi) + 71 + \sqrt{\log(1/\Delta)}) + \frac{2\bar{y} + 9\bar{F}_\alpha}{\alpha\varepsilon\sqrt{n}} + o\left( \frac{1}{\sqrt{n}} \right)$$

□

# G    HELPER RESULTS

**Proposition G.1.** *For any sample size $n$, data set $\{x_1, \cdots, x_n\}$ with size of $n$, and $\pi_1, \pi_2 \in \Pi$,*

   *1. Triangle inequality holds for $D_2(\pi_1, \pi_2) \leq D_2(\pi_1, \pi_3) + D_2(\pi_3, \pi_2)$.*

   *2. $N_2(\epsilon, \Pi, \{x_1, \cdots, x_n\}) \leq N_H(\epsilon^2, \Pi)$.*

*Proof.* Statement 1 is easy to show by triangle inequality. Statement 2 is proved similarly as in (Zhan et al., 2024, Lemma 1). □

**Proposition G.2.** *Conditioned on the data $\{X_1, \cdots, X_n\}$, the sequence of refining approximation operators $A_1, \cdots, A_J$ as constructed above satisfies the following properties:*

   *1. $\max_{\pi \in \Pi} D_2(\pi, A_J^\pi) \leq 2^{-J}$.*

   *2. $|\{A_j^\pi | \pi \in \Pi\}| \leq N_2(2^{-j}, \Pi, \{X_1, \cdots, X_n\})$, for every $j = 0, 1, \cdots, J$*

   *3. $\max_{\pi \in \Pi} D_2(A_j^\pi, A_{j+1}^\pi) \leq 2^{-(j-1)}$, for every $j = 0, 1, \cdots, J - 1$.*

4. *For any $J \geq j' \geq j \geq 0$,*

$$|\{(A_j^\pi, A_{j'}(\pi))|\pi \in \Pi\}| \leq N_2(2^{-j'}, \Pi, \{X_1, \cdots, X_n\}).$$

*Proof.* The proof can be found in (Zhou et al., 2023, Theorem 1, Step 1). $\qquad\square$

# H  FURTHER DISCUSSION AND CONCLUSIONS

In this paper, we design a risk sensitive policy learning algorithm $\lambda$-$\alpha$RSL that maximizes the weighted sum of APE and $\alpha$-level CVaR of CAPE. We show that the sample complexity of this proposed algorithm is $O(\kappa(\Pi)n^{-\frac{1}{2}})$. Numerical results show that $\lambda$-$\alpha$RSL is particularly advantageous when the objective is to improve outcomes for the worst-affected minority groups in the population, while incurring only a statistically negligible loss in overall social welfare compared to the benchmark CAIPWL, which is designed to maximize social welfare.

One possible future research direction is to design a heuristic algorithm, possibly with theoretical regret or convergence guarantee, that solves the constrained optimization problem in equation 10 proposed in Appendix B. The optimal solution to equation 10 ensures the highest attainable social welfare while simultaneously hedging against a pre-specified level of risk, making it more suitable for real-world applications.

