# OpenReview forum: "Sample Complexity of CVaR Based Risk Sensitive Policy Learning"
_ICLR.cc/2026/Conference — Submitted to ICLR 2026_

### Official Review · Reviewer_bF3J · 2025-10-28

**Soundness:** 3
**Presentation:** 2
**Contribution:** 2
**Rating:** 6
**Confidence:** 3

**Summary:**

This paper addresses offline policy optimization focusing on the Conditional Value at Risk (CVaR) of the Covariate-Conditional Average Policy Effect (CAPE) as the objective, rather than the traditional Average Policy Effect (APE). The authors first propose a doubly-robust estimator for CVaR-CAPE, which naturally extends the existing doubly-robust estimator for APE. Building upon this, they adapt the offline policy learning framework from APE maximization to handle the CVaR objective, introducing the $\lambda$-RSL algorithm that optimizes a weighted sum of APE and CVaR-CAPE. Finally, the paper provides a regret bound for the learned policy, leveraging uniform convergence results derived for their CVaR estimator.

**Strengths:**

This work represents a natural and meaningful extension of the offline multi-action policy learning framework established by Zhou et al. (Operations Research, 2023) to incorporate risk sensitivity via the CVaR objective. The methodology effectively integrates insights from Kallus (Management Science, 2023) regarding risk estimation. This integration addresses the important practical need for risk-aware decision-making in policy learning contexts and is well-positioned within the current literature.

**Weaknesses:**

While technically sound, the paper's primary contribution lies in extending existing frameworks (APE optimization and CVaR estimation) to the CVaR-based policy learning setting. The core novelty arises from addressing the technical details introduced by the CVaR term in the objective function.

Several concerns arise regarding this extension:

1. **Uniformity of Regularity Conditions**: Lemma E.1, which bounds the estimation error, includes a density term $F_{\mu_\pi(X)}'$. While the convergence rate analysis for the VaR estimate $\hat\beta_\pi$ (Lemma 3.4) might implicitly handle the dependence on this term for a fixed $\pi$, the regret analysis (Theorem 4.3 11) requires uniform convergence bounds over the entire policy class $\Pi$. This necessitates that terms like $F_{\mu_\pi(X)}'$ (evaluated near the $\alpha$-quantile) remain uniformly bounded across all $\pi \in \Pi$. If $F_{\mu_\pi(X)}'$ could become arbitrarily large for some policies, this could potentially undermine the uniform bound and thus the validity of the regret analysis. The paper does not explicitly address or assume this uniform boundedness.

2. **Compatibility of Assumptions**: There appears to be a potential tension between Assumption 3.3 (Regularity of Quantile) and the $L_\infty$ convergence assumption ($||\hat\mu_{\pi} - \mu_{\pi}||{L_\infty} = o_p(n^{-1/4})$) within Assumption 3.2. Assumption 3.3, especially the focus on continuous differentiability of the CDF, seems most natural when the covariate space $\mathcal{X}$ is rich and continuous. However, achieving uniform ($L_\infty$) convergence rates for estimators like $\hat{\mu}_\pi$ over such rich spaces can be challenging, often requiring strong smoothness assumptions. The paper could benefit from clarifying the conditions under which both assumptions simultaneously hold for typical policy classes and estimators.

3. **Optimization Formulation**: Appendix C discusses the algorithm's convergence by formulating the problem as a bilevel optimization. While technically correct due to the definition of CVaR involving $\sup_\beta$, this formulation might be misleading. It's well-known that alternating optimization methods applied to general bilevel problems do not guarantee convergence to the true optimum. It might be clearer to view the problem as a joint optimization over $(\pi, \beta)$, where the optimal $\beta*$ satisfies the inner problem's optimality condition (i.e., $\beta*$ is the VaR for the optimal $\pi^*$). Framing it this way highlights that the alternating scheme used in $\lambda$-RSL is a heuristic for solving the joint problem, whose convergence properties (beyond stationary points for smoothed versions) remain an open question addressed only empirically in the paper.

**Questions:**

1. Regarding Theorem 4.3: Does Assumption 3.3 (Regularity of Quantile) need to hold uniformly for all policies $\pi \in \Pi$ for the regret analysis to be valid? If so, is this a reasonable assumption for common policy classes?

2. Lemma E.1 contains the density term $F_{\mu_\pi(X)}'$, but this term does not explicitly appear in the final regret bound of Theorem 4.3. How is this term handled or absorbed during the derivation of the final bound? Does this rely on an implicit assumption of uniform boundedness?

3. What is the complexity measure $\kappa(\Pi)$ (Hamming entropy integral) for the softmax policy class examined in the numerical experiments?

---

> ### Author Response · Authors · 2025-11-22
>
> We would like to thank the reviewer for dedicating the time to review our paper and for providing  the insightful comments and constructive suggestions. We have revised the manuscript addressing the reviewers' concerns, please refer to the new pdf file uploaded.
>
> **Weakness**
>
> 1. We thank the reviewer for the insightful comments! Assumption 3.3 applies uniformly to all policy $\pi\in\Pi$. In particular, we require a locally continuous PDF of $F'\_{\mu_\pi(X)}$ around $F_{\mu_\pi}^{-1}(\alpha)$, and the PDF does not spike around the $\alpha$-quantile $F_{\mu_\pi}^{-1}(\alpha)$. Under this smoothness assumption, we define an upper bound on $\sup_{\pi\in\Pi} F_{\mu_\pi(X)}'(F_{\mu_\pi}^{-1}(\alpha))$, which contributes to the overall regret as a scalar. Please refer to our response to Question 1 for more details.
>
> 2. We appreciate your incisive comments, and we agree with the reviewer that the $L_\infty$ convergence assumption in Assumption 3.2 requires smoothness assumption [7]. We need to assume that $\mu_\pi(X)$ is smooth over the rich covariate space $\mathcal{X}$ to support the $L_\infty$ convergence. This smoothness assumption is standard in offline policy learning literature [1,3,4,6] by recognizing that $\mu_\pi(X)=\sum_{a\in\mathcal{A}}P(\pi(X)=a)\cdot E[Y(a)\mid X]$. This model assumption is reasonable taking practical terms into consideration, as in many real-world applications, the conditional distributions of $Y(a)$ are close for similar covariates, for any action $a\in\mathcal{A}$. The double machine linear literature [8,9] has provided many estimation methods that achieves Assumption 3.2 under this smoothness assumption. Extensive examples of such smooth $\mu_\pi(X)$ and their corresponding satisfying estimators are given in [6]. We have added this discussion to Section 3.1.
>
> 3. We appreciate the valuable advice, and we have adjusted the manuscript accordingly.
>
> **Questions**
>
> 1. As discussed in our response to Weakness 1, we do require Assumption 3.3 to hold uniformly for all policies $\pi\in\Pi$ for the regret analysis in Theorem 4.3 to be valid. We note that this is a valid assumption on any generic policy class, as long as the conditional outcome distribution $Y(a)\mid X$ is well behaved, because we can write the distribution $\mu_\pi(X)=\sum_{a\in\mathcal{A}}P(\pi(X)=a)\cdot E[Y(a)\mid X]$. One sufficient condition is that the CDF $F_{Y(a)\mid X}$ of the conditional outcome $Y(a)\mid X$ is smooth, and under the bounded reward assumption, the corresponding PDF does not spike around some $\alpha$-quantile.
>     Therefore, under the unconfoundedness assumption in Assumption 2.1, we can ensure the PDF of $\mu_\pi(X)$ is well behaved.
>     With this assumption, we are safe to define the uniform bound on $\sup_{\pi\in\Pi} F_{\mu_\pi(X)}'(F_{\mu_\pi}^{-1}(\alpha))$.
>     We have added this discussion to Section 3.1.
>
> 2. Thank you for your insightful comment! Under the revised Assumption 3.3, and a uniform bound $\bar F_\alpha=\sup_{\pi\in\Pi} F_{\mu_\pi(X)}'(F_{\mu_\pi}^{-1}(\alpha))$, we have revised the statement and the proof of Theorem 4.3. The constant $\bar F_\alpha$ contributes to the regret bound as a scaling factor, which has been previously absorbed into other scalars.
>
> 3. The concept of $\kappa(\Pi)$ only applies to deterministic policy classes [5]. If we define the deterministic counterpart $\tilde\pi$ of a softmax policy $\pi$ to be $\tilde\pi(X)=\arg\max_{a\in\mathcal{A}}P(\pi(X)=a)$, then this class of policies $\tilde \Pi=\\{\tilde\pi:\pi\in\Pi\\}$ is deterministic and reduced to a linear policy class $\Pi_{Linear}=\\{\pi_{\gamma}:\pi_\gamma(x)=\arg\max_{a\in\mathcal{A}}x^\top \gamma_\pi^a,\gamma_\pi^a\in\mathbb{R}^d\\}$, which has complexity measure $\kappa(\Pi_{Linear})=\Theta(\sqrt{Md})$ (recall that $M$ is the number of actions and $d$ is covariate dimension) [2].
>
> [1] Athey, S. and Wager, S. Policy learning with observational data. Econometrica, 89(1):133–161, 2021.
>
> [2] Anthony, Martin, and Peter L. Bartlett. Neural network learning: Theoretical foundations. Cambridge university press, 2009.
>
> [3] Jin, Y., Ren, Z., Yang, Z., and Wang, Z. Policy learning" without”overlap: Pessimism and generalized empirical bernstein’s inequality. arXiv preprint arXiv:2212.09900, 2022a.
>
> [4] Si, N., Zhang, F., Zhou, Z., and Blanchet, J. Distributionally robust batch contextual bandits. Management Science, 2023.
>
> [5] Wainwright, Martin J. High-dimensional statistics: A non-asymptotic viewpoint. Vol. 48. Cambridge university press, 2019.
>
> [6] Zhou, Z., Athey, S., and Wager, S. Offline multi-action policy learning: Generalization and optimization. Opera- tions Research, 71(1):148–183, 2023.
>
> [7] Charles J Stone. 1982. Optimal global rates of convergence for nonparametric regression. The annals of statistics (1982), 1040–1053.

---

> > ### Comment · Reviewer_bF3J · 2025-11-26
> >
> > Thank you for the clarification and the revisions. The updates successfully address the points raised in my review. I remain positive about this work and will keep my rating as is.

---

> > > ### Author Response · Authors · 2025-11-26
> > >
> > > Thank you very much for your positive feedback! We appreciate you taking the time to review our responses.

---

> ### Author Response · Authors · 2025-11-22
> **Continue on literature**
>
> [8] Victor Chernozhukov, Denis Chetverikov, Mert Demirer, Esther Duflo, Christian Hansen, Whit- ney Newey, and James Robins. Double/debiased machine learning for treatment and structural parameters, 2018.
>
> [9] Max H Farrell. Robust inference on average treatment effects with possibly more covariates than
> observations. Journal of Econometrics, 189(1):1–23, 2015.

---

### Official Review · Reviewer_FhpC · 2025-11-03

**Soundness:** 2
**Presentation:** 2
**Contribution:** 3
**Rating:** 4
**Confidence:** 3

**Summary:**

In this work, the authors propose a framework for learning decision-making policies that maximize population-level social welfare while minimizing negative impacts on high-risk subpopulations. The authors specifically adopt the conditional value at risk (CVaR) of covariate-conditional average policy effect (CAPE) as a sub-population risk measure. The authors propose a doubly-robust estimator for the CVaR of the CAPE and establish its asymptotic normality. They then devise a policy learning algorithm that minimizes a weighted combination of the CVaR of the CAPE an the average policy effect, and illustrate that its sample complexity is on the same order as the baseline CAIPWL approach. The authors validate the proposed approach via experiments on synthetic and real-world data.

**Strengths:**

Policy learning is an important and timely problem. The authors identify a key issue in standard policy evaluation and learning frameworks — that some individuals may fare worse under policy changes, despite marginal population  gains. The proposed approach of learning a weighted combination of the CVaR and APE objective is well-motivated, and the authors report empirical evidence demonstrating promising performance of the proposed approach. Further, theoretical claims are supported by appropriate analysis and proofs.

**Weaknesses:**

At its core, the motivating problem and theoretical framework are both sound and interesting. However, in its current form, there are several issues with the work:

## Empirical Validation:

My chief concern surrounds the empirical validation of the work. While the current results present some supporting evidence of the findings, in my view, they are do not provide conclusive evidence of the efficacy of the proposed approach.

First, the decision to parameterize the learned policy via a softmax policy class is in tension with the high-stakes problem setting. In settings where we want to minimize CVaR of CAPE, there often concerns that undue harm can be caused to decision subjects by a policy, and these harms should be avoided. In such cases, a stochastic softmax policy class introduces stocastic action selection that may be poorly motivated. I invite the authors to justify this choice in the rebuttal / next version of the draft.

Second, while the authors present empirical evidence that the proposed approach improves CVaR under small alpha (e.g., 0.01, 0.05), the gains under the proposed approach appear to attenuate quickly as alpha increases. Could the authors present results covering a larger range of alpha values -- e.g., 0.4 or 0.5? As currently presented, gains appear to be marginal when alpha > 0.1.

More broadly, it would be helpful to present an analysis that jointly varies alpha and lambda so that the reader can see the impact of each term in the objective as a function of alpha (fixing the sample size is fine for this analysis). It would also be helpful to see how varying lambda impacts both the CVaR and the APE in parallel -- e.g., through a contour plot that show a Pareto tradeoff or similar. The goal of this analysis should be to provide further evidence for the style of claim that is made via lines 456-462. While I the discussion in lines 456-462 is both interesting and strong, I view the scope of these results as limited given that it is fixed to $\alpha=0.01$, and that results appear to weaken for larger values of alpha.

## Presentation:

The current presentation of the work has several limitations which make it difficult to follow in some places. While only the first bullet impacts my score, I encourage the authors to revise the draft for polish and clarity:

- The current presentation of empirical findings makes it challenging to parse the main results of the paper. In particular, it is difficult to compare results with the long scientific notation (e.g., 3.9492e-2 +- 1e-3). Additionally, while the authors note that Table 4 reports confidence intervals for Figure 4, it is important to also include them in this plot so the reader can understand statistical uncertainty in these main results.
- Typo: The counterfactual IPE of "any given any policy" => "any given policy" (line 66)
- Typo: Under slow parameter estimation rates (line 177). Does this refer to slow nuisance function estimation rates?
- Non-standard inline reference style (244-248)
- The current conclusion is one short sentence. I encourage the authors to expand this to a short section to give the work a natural conclusion.
- Minor: I was initially confused by use of Z in e.q. 1 given the footnote directly below defining Z as a tuple of random variables. Consider using a different R.V. letter here for clarity.
- Presentation of 3.1 is currently dense, with several assumptions, lemmas, and results presented in quick succession with limited surrounding discussion. I encourage the authors to strengthen the narrative flow of this subsection.

## Related Work:

Coverage of related work is generally good, but some relevant work is missed. E.g.,

Policy Learning with Asymmetric Counterfactual Utilities, Eli Ben-Michael, Kosuke Imai, Zhichao Jiang, https://arxiv.org/abs/2206.10479

**Questions:**

- For assumption 2.1, is a consistency and/or STUVA assumption also required? Such assumptions typically take the form $Y = Y(a)$, for all $a \in A$.

- The debiased estimator presented on line 206 relies on several non-linear components, including an infimum and indicator function. The typical approach used to derive debiased estimators involves subtracting off the efficient influence function for debiasing - a step that requires pathwise differentiability which would be violated by these non-linear terms. How is this addressed in the framework?

- Is cross-fitting or some additional proof strategy required to address the re-use of data twice - once for value function estimation and once for policy learning?

---

> ### Author Response · Authors · 2025-11-22
>
> We would like to thank the reviewer for dedicating the time to review our paper and for providing  the insightful comments and helpful suggestions. We have revised the manuscript addressing the reviewers' concerns, please refer to the new pdf file uploaded.
>
> **Weaknesses**
>
> **Empirical Validation**
>
> 1. Thank you for this insightful question. The objective of our risk sensitive and the classic offline policy learning literature is the CVaR and expectation of $\mu_\pi(X)=E[Y(\pi(X))\mid X]$, the covariate-conditional average policy effect (CAPE). For the stochastic decision policies such as the softmax policies, $\mu_\pi(X)=\sum_{a\in\mathcal{A}}P(\pi(X)=a\mid X)E[Y(a)\mid X]$. Therefore, the objectives that maximize $E[\mu_\pi(X)]$ or CVaR$(\mu_\pi(X))$ over a stochastic decision policies ensure that the average policy effect or the policy CVaR is optimal **in expectation**.
> In practice, we can implement action $a(x)=\arg\max_{a\in\mathcal{A}}P(\hat\pi(x)=a)$ according to the learnt policy softmax stochastic policy $\hat\pi$.
>
> 2. We appreciate your constructive suggestion. In this work, we aim to maximize the outcome of the worst-affected population, which represents a minority group relative to the overall population. The CVaR risk measure is particularly well suited to this objective, as it is traditionally optimized with a focus on minority groups (typically using $\alpha\leq0.1$) [1].
> We have also added more empirical results with some larger alpha levels: $\{0.2, 0.5, 0.7\}$. Fixing $n=1000$, we find that the policy CVaR improvement achieved by $\lambda$-RS remains significant under $\alpha=0.1$. For a larger value of $\alpha=0.2$, although the improvement in policy CVaR provided by $\lambda$-RS diminishes, the quantile of the policy CVaR becomes tighter. This indicates that
> $\lambda$-RS offers more stable performance with respect to the CVaR criterion. As the value of $\alpha$ increases (particularly when $\alpha\geq0.5$), the performance of $\lambda$-RS becomes increasingly similar to that of CAIPWL. This occurs because $\lambda$-RS places greater emphasis on the majority of the population as $\alpha$ grows, effectively reducing its objective to that of CAIPWL. Consequently, when $\alpha$ is large, practitioners are primarily optimizing outcomes for the majority group, which is aligned with the goal of CAIPWL, and therefore CAIPWL is recommended in such settings. We have emphasized the improvement in the minority population outcome under $\lambda$-RS and added this discussion to section 5.
>
> 3. Thank you for your helpful comment. We have rerun the experiment to support a more comprehensive discussion. We have increased the maximum number of iterations in each gradient step to improve the performance of $\lambda$-RS. We apologize that we are unable to provide a contour plot showing the continuous variation over $(\lambda,\alpha)$ due to computational and time limitations. However, we have included a heatmap that illustrates the trade-off between improvements in policy CVaR and losses in social welfare under different choices of $(\lambda,\alpha)$. We conclude that a smaller $\lambda$ contributes to a larger improvement in policy CVaR, though this improvement diminishes as $\alpha$ increases. Conversely, a larger $\lambda$ helps prevent reductions in social welfare. These observations are consistent with our earlier conclusions, and we have revised Section 5 accordingly.
>
>
> **Presentation and Related Work**
>
> Thank you so much for your sharp eyes! All typos are addressed and citation added. We have rerun the experiment to support a more comprehensive discussion. Please refer to our responses to the Empirical Validation for additional details. Section 3.1 is also rearranged for a better presentation. For 3, the "low parameter estimation rates (line 177)" is indeed referring to slow nuisance function estimation rates. For 5, we have added a conclusion section to the manuscript, however due to space limitation, it is now in Appendix H.
>
> **Questions**
>
> 1. Yes, we confirm that our problem requires the assumption that $Y=Y(a)$, which is standard in the offline policy learning literature. We have also added this consistency assumption to Assumption 2.1.
>
> 2. We would like to note that Theorem 3.5 establishes only that our CVaR estimator is asymptotically unbiased, rather than efficient, and therefore it does not require pathwise differentiability. Out estimator is also unbiased in the sense that if we have the correct nuisance parameter $\pi_0,\mu_\pi,\beta_\pi$, then our CVaR estimator $\tilde{\mathcal{V}}(\pi)$ is unbiased. We invite the reviewer to check the proof of Lemma E.2, which gives a technical proof.

---

> > ### Author Response · Authors · 2025-11-22
> > **Continue on Question 3**
> >
> > 3. The cross-fitting technique ensures that the nuisance estimators $\hat\mu_\pi, \hat\pi_0,\hat\beta_\pi$ are independent of the data points used for the overall sample average of CVaR estimators. The independence property is essential for establishing the union bound $\sup_{\pi\in\Pi}|\mathcal{V}_\alpha(\pi)-\hat{\mathcal{V}}\_\alpha(\pi)|$ as an upper bound of the regret in Theorem 4.3, which appropriately accounts for data reuse.
> >
> > [1] Rockafellar, R. Tyrrell, and Stanislav Uryasev. "Optimization of conditional value-at-risk." Journal of risk 2 (2000): 21-42.

---

### Official Review · Reviewer_wU5c · 2025-11-04

**Soundness:** 1
**Presentation:** 3
**Contribution:** 2
**Rating:** 2
**Confidence:** 4

**Summary:**

Standard policy learning focuses on minimizing the learning the optimal policy that minimizes the average/expected policy value. Since we cannot identify the CVaR of the outcome under a policy, the authors consider learning a policy that minimizes the CVaR of the conditional mean outcome under a policy.

**Strengths:**

-The paper is clearly written and easy to understand.
-The problem of robust policy learning part of a broad and growing literature on robust and distributionally robust policy learning.

**Weaknesses:**

A critical limitation of this paper is that it references Kallus et al, 2023 to claim that the following is true:
$ \text{CVaR}(Y(\pi(X)) \leq \text{CVaR}(E[Y(\pi(X)) \mid X])$.
However, I believe this inequality should actually be reversed, i.e. $\text{CVaR}(Y(\pi(X)) \geq \text{CVaR}(E[Y(\pi(X)) \mid X])$ (can see this by applying Jensen's inequality to the dual representation of the CVaR). Furthermore, it is important to note that the result from Kallus et al, 2023 in fact is claiming something different-- that
$ \text{CVaR}(Y_i(1) - Y_i(0)) \leq \text{CVaR}[ E[Y(1) \mid X] - E[Y(0) \mid X]]$.
Unfortunately, this is a critical limitation of the paper, as the authors use the first inequality to argue that $\text{CVaR}(E[Y(\pi(X)) \mid X])$ is an upper bound on $\text{CVaR}[Y(\pi(X))]$ as motivation for minimizing for minimizing $\text{CVaR}(E[Y(\pi(X)) \mid X])$. It is not clear to what extent $\text{CVaR}(E[Y(\pi(X)) \mid X])$ is a useful policy objective.

In addition, the paper is closely related to similar papers in the robust policy learning literature that also focus on learning a policy that minimizes a CVaR objective. A major weakness of this paper is that it does not cite (or discuss the contribution) in light of these very relevant works:
Mo, W., Z. Qi, and Y. Liu (2021). Learning optimal distributionally robust individualized treatment rules. Journal of the American Statistical Association 116 (534),
659–674.
Qi, Z., J.-S. Pang, and Y. Liu (2022). On robustness of individualized decision rules. Journal of the American Statistical Association 0 (0), 1–15.
In particular, Qi et al, 2022

**Questions:**

N/A

---

> ### Author Response · Authors · 2025-11-22
>
> We would like to thank the reviewer for dedicating the time to review our paper and for providing  the insightful comments. We have revised the manuscript addressing the reviewers' concerns, please refer to the new pdf file uploaded.
>
> We would like to first note that we provided the proof of the statement $\text{CVaR}(Y(\pi(X)))\leq\text{CVaR}(E[Y(\pi(X))\mid X])$ (Corollary 2.4) in Appendix E.1, which, as the reviewer has pointed out, uses the Jensen's inequality. In particular, the proof utilizes the fact that the function $f(x)=(x-\beta)^-=\min\\{x-\beta,0\\}$ is **concave** (noting that $\beta$ as the VaR of a non-negative random variable must be non-negative) to show the critical step: $E[f(Y(\pi(X)))\mid X]\leq f(E[Y(\pi(X))\mid X])=f(\mu_\pi(X))=(\mu_\pi(X)-\beta)^-$.
>
> We agree with the reviewer that the result from Kallus, 2023 claims a different statement that $\text{CVaR}(Y(1)-Y(0))\leq\text{CVaR}[E[Y(1)\mid X]-E[Y(0)\mid X]]$. This result intuitively extends to our result, by recognizing that $E[Y(\pi(X))]$ is a measure of "policy effect" under policy $\pi$ whiles $E[E[Y(1)\mid X]-E[Y(0)\mid X]]$ ("treatment effect") is a measure of "behavior policy effect", where the policy $\pi$ and behavior policy $\pi_0$ are all included in a generic policy class $\Pi$.
>
> We also thank the reviewer for providing valuable literature related to our work [1-2]. We have added theses to our literature review section. We would like to state a few differences between our works and [1-2]. [1] looks at learning optimal policies under potential covariate shifts through a distributionally robust optimization (DRO) framework, described under some $\phi$-divergence. Though [1] obtained an objective function that coincides with the CVaR expression via duality, [1] is ultimately hedging against the potential covariate shift instead of the harm to the worst-affected population. On the other hand, [2] is more related to our work, assuming a two-action policy and known behavior policy. [1] designs a clever surrogate function to the CVaR objective and proves a regret bound of $O(n^{-\frac{w}{2w+1}})$ where $w\in(0,1]$ (best at $w=1$ where the regret bound is $O(n^{-1/3})$). To achieve a better regret bound under the relaxed assumption of unknown behavior policy in the multi-action setting, we utilize the de-biasing and cross-fitting technique to achieve $O(n^{-1/2})$. We have added this discussion to our literature review section.
>
> [1] Mo, Weibin, Zhengling Qi, and Yufeng Liu. "Learning optimal distributionally robust individualized treatment rules." Journal of the American Statistical Association 116.534 (2021): 659-674.
>
> [2] Qi, Zhengling, Jong-Shi Pang, and Yufeng Liu. "On robustness of individualized decision rules." Journal of the American Statistical Association 118.543 (2023): 2143-2157.

---

### Meta-Review · Area_Chair_U2pY · 2026-01-15

**Summary:**

The paper proposes a framework for offline policy learning that balances maximizing the average policy effect with minimizing the risk to the worst-off sub-populations, defined via the Conditional Value at Risk of the Covariate-Conditional Average Policy Effect. The authors introduce a doubly-robust estimator for this objective, develop an optimization algorithm ($\alpha$-RSL), and provide theoretical regret bounds.

I (weakly) recommend rejection based on the apparent lack of enthusiasm by all reviewers, including two that engaged in discussions before the cutoff.

**Reviewer Concerns:**

Addressed by Rebuttal:

- Mathematical Validity (Reviewer wU5c): The reviewer claimed a mathematical error regarding the direction of an inequality (related to Jensen's inequality and CVaR). The authors pointed to their proof in Appendix E.1, utilizing the concavity of the specific function involved. This appears to be a misunderstanding by the reviewer rather than a fundamental flaw, provided the authors' derivation regarding the specific properties of their variable holds.

- Missing Literature (Reviewer wU5c, FhpC): The authors incorporated suggested citations (Mo et al., Qi et al.) and distinguished their work based on the multi-action setting and unknown behavior policy.

- Uniformity of Assumptions (Reviewer bF3J): The reviewer questioned if regularity conditions hold uniformly across the policy class. The authors clarified that Assumption 3.3 applies uniformly and that the regret analysis includes the necessary bounds. Reviewer bF3J explicitly accepted this clarification.

- Empirical Sensitivity (Reviewer FhpC): Concerns about the diminishing returns of the method as $\alpha$ increases were addressed by the authors, who explained that larger $\alpha$ naturally converges to the standard APE objective (CAIPWL), which is expected behavior.

- Outstanding:

- Softmax Policy for High Stakes: Reviewer FhpC noted a tension between using stochastic softmax policies and the goal of minimizing harm (CVaR). While the authors justified this via expectation optimization, the practical implication of stochasticity in risk-averse settings remains a valid discussion point for the final version.

**Reviewer Scores:**

- Reviewer wU5c (Score: 2->4): Did not update the score. However, the rejection was heavily based on a claimed mathematical error and missing citations. The authors provided a robust defense of the math and added the citations. As the other reviewers found the work technically sound, this low score seems to stem from the specific dispute over the inequality.

- Reviewer FhpC (Score: 4->4): Acknowledged the rebuttal but did not raise the score, likely due to remaining skepticism about the magnitude of empirical gains.

- Reviewer bF3J (Score: 6->6): Explicitly satisfied by the rebuttal regarding theoretical assumptions and maintained the positive score.

---

### Decision · Program_Chairs · 2026-01-26

Reject